# Recognition and reconstruction of cell differentiation patterns with deep learning

**Robin Dirk**[�], **Jonas L. Fischer**[�], **Simon Schardt**[�], **Markus J. Ankenbrand, Sabine C. Fischer**[iD]*

Julius-Maximilians-Universität Würzburg, Fakultät für Biologie, Center for Computational and Theoretical Biology, Klara-Oppenheimer-Weg 32, Campus Hubland Nord, Germany

☯ These authors contributed equally to this work.

* sabine.fischer@uni-wuerzburg.de

**Data Availability Statement:** All machine learning models were constructed in TensorFlow for Python 3. Code and data sets are available on our github

## Abstract

Cell lineage decisions occur in three-dimensional spatial patterns that are difficult to identify by eye. There is an ongoing effort to replicate such patterns using mathematical modeling. One approach uses long ranging cell-cell communication to replicate common spatial arrangements like checkerboard and engulfing patterns. In this model, the cell-cell communication has been implemented as a signal that disperses throughout the tissue. On the other hand, machine learning models have been developed for pattern recognition and pattern reconstruction tasks. We combined synthetic data generated by the mathematical model with spatial summary statistics and deep learning algorithms to recognize and reconstruct cell fate patterns in organoids of mouse embryonic stem cells. Application of Moran's index and pair correlation functions for *in vitro* and synthetic data from the model showed local clustering and radial segregation. To assess the patterns as a whole, a graph neural network was developed and trained on synthetic data from the model. Application to *in vitro* data predicted a low signal dispersion value. To test this result, we implemented a multilayer perceptron for the prediction of a given cell fate based on the fates of the neighboring cells. The results show a 70% accuracy of cell fate imputation based on the nine nearest neighbors of a cell. Overall, our approach combines deep learning with mathematical modeling to link cell fate patterns with potential underlying mechanisms.

## Author summary

Mammalian embryo development relies on organized differentiation of stem cells into different lineages. Particularly at the early stages of embryogenesis, cells of different fates form three-dimensional spatial patterns that are difficult to identify by eye. Pattern quantification and mathematical modeling have produced first insights into potential mechanisms for the cell fate arrangements. However, these approaches have relied on classifications of the patterns such as inside-out or random, or used summary statistics such as pair correlation functions or cluster radii. Deep neural networks allow characterizing patterns directly. Since the tissue context can be readily reproduced by a graph, we implemented a graph neural network to characterize the patterns of embryonic stem cell

page (https://github.com/scfischer/dirk-et-al-2022) and archived at Zenodo with doi: 10.5281/zenodo. 7458420.

**Funding:** SCF acknowledges funding from the Deutsche Forschungsgemeinschaft (DFG, German Research Foundation) project number 470129398 and start-up funding RAC/2020 from the University of Wuerzburg. The funders had no role in study design, data collection and analysis, decision to publish, or preparation of the manuscript.

**Competing interests:** The authors have declared that no competing interests exist.

organoids as a whole. In addition, we implemented a multilayer perceptron model to reconstruct the fate of a given cell based on its neighbors. To train and test the models, we used synthetic data generated by our mathematical model for cell-cell communication. This interplay of deep learning and mathematical modeling in combination with summary statistics allowed us to identify a potential mechanism for cell fate determination in mouse embryonic stem cells. Our results agree with a mechanism with a dispersion of the intercellular signal that links a cell's fate to those of the local neighborhood.

## Introduction

During mammalian embryogenesis, multilineage primed stem cells differentiate into functionally diverse cell types. While the population of stem cells displays a large heterogeneity due to the underlying molecular mechanisms, lineage decisions occur in ordered three-dimensional spatial cell fate patterns [1, 2]. Mouse embryonic stem cells (mESCs) possess the ability to generate three-dimensional organoids, self-organize and differentiate into embryonic and extra-embryonic precursor cells comparable to the inner cell mass (ICM) of the mouse blastocyst [3–5].

Different approaches have been developed for describing the cell fate patterns in mESC cultures. For visually identifiable patterns such as an engulfing pattern with cells of one fate surrounded by cells of a different fate, a manual classification system has been implemented [6]. Patterns that are visually more challenging have been characterized based on their clustering characteristics, such as the number and size of clusters [7, 8], as well as the cell fate composition of the local neighborhood [2, 5, 8]. Subsequent comparisons with random or simple rule-based patterns have emphasized the complex nature of the cell differentiation patterns.

Machine learning methods have been applied to many problems in the broad field of pattern recognition. Perhaps the most popular problem that machine learning methods are known for, image recognition, is a pattern recognition problem in itself [9]. In bioinformatics, pattern recognition through machine learning models has been employed e.g. for the classification and clustering of -omics data [10] including spatial transcriptomics [11]. However, studies using machine learning for characterization of patterns arising during cell differentiation are remarkably scarce.

Therefore, we asked these questions: Can we use machine learning methods to characterize the visually unrecognizable cell differentiation patterns and how do they compare to summary statistics approaches? And, can machine learning be used to predict individual cell fates based on the surrounding cells?

For the first question, we calculated pair correlation functions and Moran's indices for experimental and synthetic data generated by our cell differentiation model [12]. The model focuses on cell fate pattern generation due to a distance-based intercellular signaling mechanism. Depending on the dispersion of the signal, patterns varying from checkerboard-like to engulfing arise. Furthermore, we implemented a graph neural network (GNN) and trained and tested it on the synthetic data. The GNN essentially addresses the inverse problem. It was trained to predict the dispersion range for a given input pattern. Subsequent application of the GNN to the data from our ICM organoids [5] suggests that the patterns are consistent with a mechanism that is based on short range cell communication, i.e. low signal dispersion, while pair correlation functions and Moran's indices suggested interactions at different ranges. This lead to the second question. Provided that cell-cell communication occurs only at short ranges, a cell's neighborhood should hold enough information to infer its fate. Therefore, we

**Table 1. Data sets used in this study.**

| ID | Data sets | Dimension | Number of replicates | Format of q | Pattern recognition | Pattern reconstruction |
|----|-----------|-----------|----------------------|-------------|---------------------|------------------------|
| A | simulated | 2D | 1427 | continuous | x | |
| B | simulated | 3D | 3287 | continuous | x | |
| C | simulated | 2D | 900 | discrete | | x |
| D | simulated | 3D | 900 | discrete | | **x** |
| E | experimental | 3D | 76 | — | **x** | **x** |

investigated whether the cell fate pattern of an organoid can be reconstructed by spatial imputation of the fate of a given cell based on the fates of its neighbors using a multilayer perceptron model. After model development and validation on synthetic data, we applied it to the ICM organoid data. We find that indeed the fate of a cell can be predicted with 70% accuracy based on the fates of its nine nearest neighbors.

## Materials and methods

### Data for pattern recognition and pattern reconstruction

In this study, we integrated simulated and experimental data sets to train and test our machine learning models for pattern recognition and pattern reconstruction (Table 1).

**Simulated data.** We employed our previously established model for cell signaling to generate in silico tissues with different patterns ranging from checkerboard-like to engulfing [12] (see also S1 Text for further details). In this agent-based model, each cell is defined by the position of its centroid as well as the expression of two transcription factors $u$ and $v$. The cell neighborhood relationships are described by a Delaunay cell graph [13]. Initially, all cells have similar levels of $u$ and $v$. A combination of intracellular mutual inhibition and intercellular signalling leads to differential expression of u and v. The resulting cells are classified as u+v− (u high / v low) or u−v+ (u low / v high) cells, which we simplified to 0 and 1. A dispersion parameter $q$, with values between 0 and 1 regulates how strongly the intercellular signal propagates in the in silico tissue. Controlling $q$ allows a variety of different patterns to be generated by the two cell types. For $q$ close to zero, cells interact mainly with their direct neighbors. This yields a checkerboard-like pattern in which adjacent cells adopt different fates as much as the geometry allows. Increasing the dispersion parameter $q$ ensures that cells that are further away are more strongly involved in a cell's fate decision. This yields more locally clustered and radially distributed cell types. For $q$ close to 1, we obtain a pattern in which one cell type is completely engulfing the other (Fig 1).

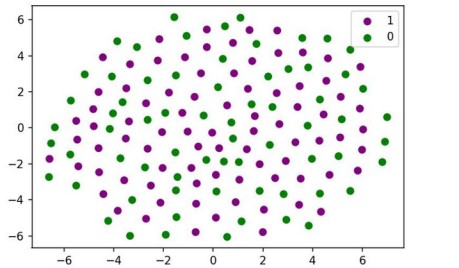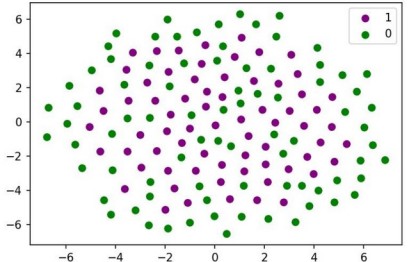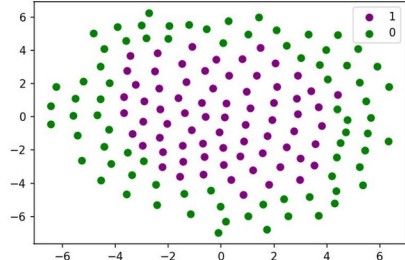

**Fig 1. Example visualizations of simulated 2D colonies with different dispersion parameters 0.2, 0.6, 0.9.**

With this model, four data sets were generated (Table 1, data sets A-D), two for pattern recognition and two for pattern reconstruction. The simulated data consists of distinct data sets with unique IDs and dispersion parameters, the position of each cell in two or three spatial dimensions (2D: CentroidX, CentroidY; 3D: CentroidX, CentroidY, CentroidZ) and the cell fate 0 and 1. A detailed list of parameters and their values can be found in S1 Text.

For pattern recognition, first, we performed simulations in two spatial dimensions resulting in 4000 2D colonies with a fixed cell count of 150 cells. To obtain this, we chose a maximum number of 6000 time steps for tissue growth with a simulation time of $T = 30$ and stopped the simulations as soon as the 150th cell arose. Afterwards, the resulting 2D colonies were used in the cell fate decision model from [12] to generate various patterns based on random values of $q \in (0, 1)$. To this end, we used a combination of 1000 time steps and a simulation time of $T = 100$. Our goal was the recognition of a binary spatial pattern. Therefore, we focused on colonies that had a sufficient number of both cells types. We excluded 2573 colonies since they contained more than 2/3 of one cell type. This threshold is motivated by the final cell fate ratio of primitive endoderm to epiblast cells in the mouse embryo of 2/3 to 1/3 [14]. The resulting data contained 1427 2D colonies (data set A). Also, we performed simulations in three spatial dimensions, resulting in 10.000 organoids with sizes between 612 and 1401 cells. These varying sizes were used to capture the biological variation in the experimental data. To achieve this, we calculated the mean number of cells from the experimental data for 24 h and 48 h organoids (441.73 and 1041.24, respectively) and determined the simulation time for the tissue growth simulation to reach these cell numbers (23.09 and 27.06, respectively). Afterwards, the cell fate decision model was used with the same parameters as in the two-dimensional simulations. Here, 6713 organoids with unbalanced cell fates were excluded. The resulting data set consisted of 3287 organoids with dispersion parameters between $6 * 10^{-5}$ and 0.999 (data set B).

For pattern reconstruction, 100 2D colonies and organoids each were simulated for the discrete dispersion parameter values $q = 0.1, 0.2, \ldots, 0.9$ over a simulation time of $T = 25$ for 6000 time steps, combined with the cell fate decisions model as before (data sets C and D). The data created in two spatial dimensions contained 900 colonies with an average size of 132 cells (data set C). The performed simulations in three spatial dimensions resulted in 900 organoids with an average size of 330 cells (data set D). The differences in cell number occur due to the differences in reduction of the cell radius upon division with

$$r_{new} = 2^{-1/dim} r_{old}, \qquad dim \in \{2, 3\}.$$

The colonies and organoids in both simulated data sets showed a balanced cell fate ratio. Therefore, no replicates had to be excluded.

**Experimental data.** We used our previously published experimental data for ICM organoids [5]. The data has been obtained by segmentation of three-dimensional confocal image stacks with fluorescently labeled cell nuclei and consists of the organoid ID, the stage (24 h or 48 h), the batch, the centroid of the cell nucleus in three dimensions (CentroidX, CentroidY, CentroidZ), and the average NANOG and GATA6 expression levels per cell nucleus.

For pattern recognition, a binary cell fate was needed. For the spatial statistics analyses, we started with the four cell fates high NANOG and high GATA6 levels (N+G+), low NANOG and low GATA6 levels (N−G−) as well as high NANOG and low GATA6 levels (N+G−) and low NANOG and high GATA6 levels (N−G+). This is the typical classification in mouse developmental research. The classes were assigned as previously described [5]. While N+G− and N−G+ cells are considered to be distinct cell precursors, N+G+ cells represent cells which are not yet committed to one of the cell fates. Contrary to this, N−G− cells are likely representing

late stage Epi and PrE cells, that have started to downregulate NANOG and GATA6 expressions. We assigned N−G− and N+G+ cell fates randomly to N+G− or N−G+. The probability to assign a cell with N+G− fate equals the sum of all N+G− cells divided by the sum of all N+G− and N−G+ cells. This ensures that the cell type proportions will not be changed by this assignment. This process is repeated 1000 times.

For the application of the graph neural network, all cells were classified into two clusters based on the GATA6 and NANOG expression levels. We took the logarithm of the GATA6 and NANOG expression values and applied a k-means clustering with a target of two clusters and random initial centroids. Cells in the cluster with high GATA6 expression levels were labeled as 0 and those in the cluster with high NANOG values as 1.

For pattern reconstruction, we considered the four cell fates high NANOG and high GATA6 levels (N+G+), low NANOG and low GATA6 levels (N−G−) as well as high NANOG and low GATA6 levels (N+G−) and low NANOG and high GATA6 levels (N−G+), which were assigned according to [5].

**Data conversion to graph structure.**   For pattern recognition with spatial statistics and the graph neural network, we converted the data from the simulations as well as *in vitro* experiments into a graph using the NetworkX package version 2.8.5 [15]. First, the Delaunay triangulation of cells as points was created. Between two cells, an edge was created if they were connected in the Delaunay graph and the distance of one cell to the other was lower than a predefined threshold. This threshold was set to two for simulated colonies and organoids, as this represents double the maximum possible cell radius in the computational model. For experimental organoids from data set E, the mean of all distances of all cells to their neighbors plus two times its standard deviation was taken as threshold. The distances between the cells are assigned as the edge weights. The distances were normalized before being assigned.

## Spatial statistics

Cell differentiation patterns in our case are the result of two different cell types arising in a tissue. To quantify spatial auto-correlation of the cell fate patterns, we used Moran's index [16]. For a variable of interest $x \in \mathbb{R}^n$, it is defined as

$$I := \frac{n}{\sum_{i,j} A_{i,j}} \frac{(x - \bar{x})^T A (x - \bar{x})}{(x - \bar{x})^T (x - \bar{x})},$$

where $\bar{x}$ denotes the mean of $x$ and $A$ is a matrix representation of the connectivity of the $n$ cells in ICM organoids or the simulated 2D colonies and organoids. For $A$, we chose the adjacency matrix of our Delaunay cell graph. The two fates of the cells are represented by $x$, a vector with entries $x_i \in \{0, 1\}$ for $i = 1, \ldots, n$. The value of $I$ is independent on which value is assigned to which fate. A value of $I$ close to -1 means that the neighboring cells are of opposite type (checkerboard or period two pattern). If cell fates form large clusters, Moran's index will be positive, and it tends to 1 for a clear separation of two cell fates.

Patterns with two cell types have also been quantified using pair correlation functions (PCFs) depending on the cell distances $d_{ij}$ [12]. The PCF relates the number of cell pairs of equal type to the random chance of picking two cells of equal type. Thus, it enables us to find accumulations of cell types at given distances towards each other. For a uniformly distributed amount of the two cells, the correlation function returns a value close to 1 for every cell distance. Consequently, deviations from 1 yield information about how much more or fewer equal cell pairs are found in certain ranges (see S2 Text for further details).

## Neural network for pattern recognition

For pattern recognition, we implemented a machine learning approach which predicts the dispersion parameter $q$ for each 2D colony or organoid. We systematically tested multilayer perceptron models with several hyperparameters on the simulated data. Specifically, different number of nodes, number of hidden layers, learning rate and activation functions were tested. It was found that multilayer perceptron models (MLP) did not perform well. We hypothesize that the relationship between data points is hardly trainable in an MLP and thus it cannot well account for the actual spatial relationships of cells in organoids. As the data is also unordered for every organoid, the "index-based" relationships cannot be learned in a way such as in image classification, where a unique pixel is represented by a datapoint at the same index for every image. Thus, we decided to consider different approaches. The organoids are coherent tissue structures, which we described by a Delaunay cell graph [13]. To enable direct input of cell neighborhood information into our machine learning methods, we employed graph neural networks (GNNs).

**Model architecture.** In GNNs, convolutional layers can be used to employ a function over a node and its neighbors, similar to how functions can be employed over neighboring pixels in convolutional networks for image recognition. This way, recognition of both local and global features is possible. For the implementation of the GNNs, we used the Python3 spektral package version 1.2.0 [17, 18]. The model architecture was inspired by examples in spektral documentation and experimentally altered to find the best performing model. The GNN for prediction of the dispersion parameter from whole simulated 2D colonies or organoids will be referred to as Model1 in the following.

Model1 consists of two convolutional layers with 450 and 150 channels. The number of channels defines the size of the matrix of learnable weights. The convolutional layers are GCSConv layers as implemented in the spektral package. They compute

$$Z' = D^{-1/2}AD^{-1/2}XW_1 + XW_2 + b$$

where $Z'$ is a matrix of node features passed to the next layer, with the last dimension changed to the number of channels. Thus, $Z'$ has the dimensions $(n, C)$ where $n$ is the number of graph nodes and $C$ is the number of defined channels. $D$ is the matrix of node degrees, $A$ is the adjacency matrix, $X$ is the node feature matrix, $W$ is a trainable weight matrix, and $b$ is a trainable bias vector. The convolutional layers are followed by a global pooling layer (Spektral GlobalAttnSumPool). Two densely connected layers follow, with 50 and 1 node(s). Between the two dense layers, a further layer is used for flattening, as we only need one value as prediction output. The architecture of Model1 both for 2D and 3D data is schematically shown in Fig 2.

**Training.** A subclass of the spektral data set class was created to store and later pass the graphs for training. Here, the graph is contained as a sparse matrix representing source and destination nodes along the respective edge weights. Furthermore, the data set object contains the cell fates as node features and the dispersion parameter for each 2D colony or organoid as graph label.

Following a data split into training (80%), testing (10%) and validation (10%) data, spektral data loaders were created from the data set objects. The loaders chosen were DisjointLoaders as implemented in spektral. Here, batches of graphs with configurable size are used to pass the data for training and testing. In DisjointLoaders, a batch of graphs is represented by their disjoint union. Graphs in one batch are not connected to each other by any edge, however share the same adjacency matrix. For each node, the index of the graph it originated from is saved. Using the index, the graphs can later be pooled independently from each other.

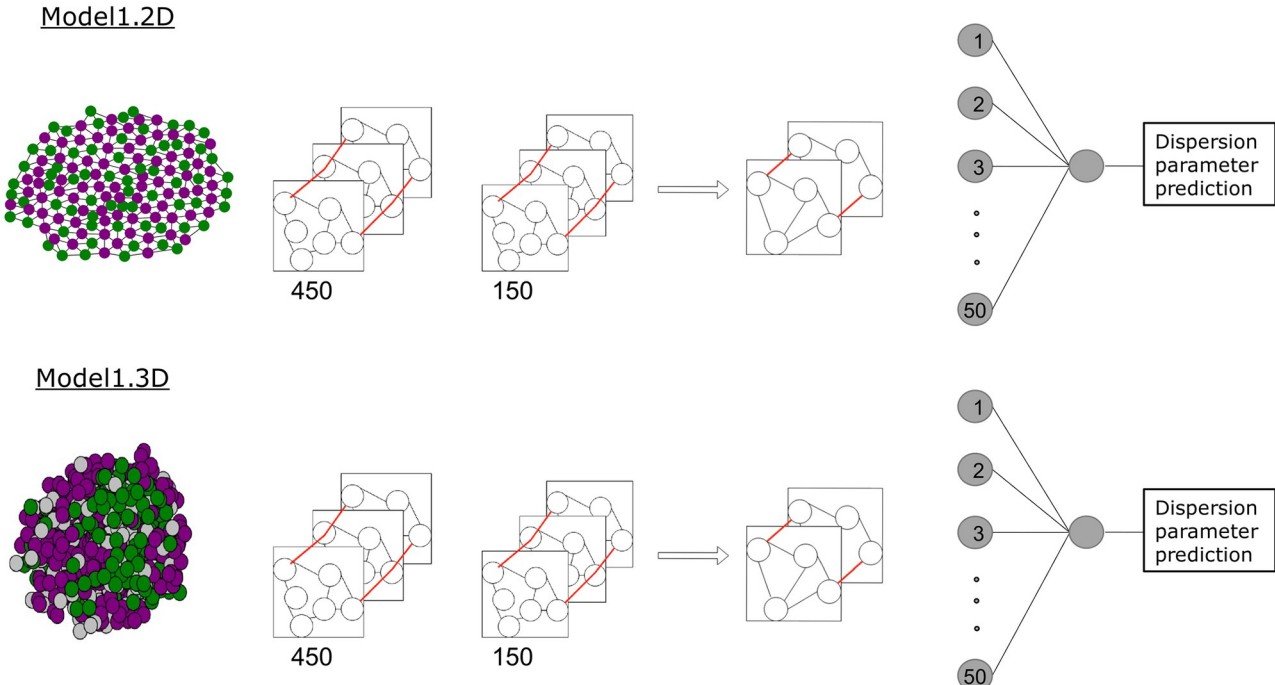

**Fig 2. The model architecture of Model1.** The input is passed as a graph, representing the cells of a two-dimensional colony or a three-dimensional organoid. Two GSCConv layers, a pooling layer and two densely connected layers follow. In the graph convolutional and dense layers, learnable weights are trained with labeled training data. The last layer consists of a single node, which represents the prediction of the dispersion parameter.

Graphs and loaders were created both for data set A and B. With the data from the training loaders, Model1 was trained. In the following, we will refer to the model trained with 2D colonies from data set A as Model1.2D and to the model trained with 3D organoid data from data set B as Model1.3D.

As training labels, we used the dispersion parameter $q$ from our simulations. Hence, $q$ was the value that was predicted by the model. Since this is a regression problem, the mean squared error (MSE) was chosen as the value to optimize ("loss"). For optimization, the Adam optimizer was used. We implemented an adaptive learning rate such that the starting learning rate was 0.0001, and it was lowered by a factor of 0.2 after ten epochs of no improvement. After settling for the described architecture, hyperparameters like number of channels, learning rate and activation functions were systematically tested. We tested every combination of 50, 150, 450 channels, "sigmoid" and "relu" activation function in every layer and 0.001, 0.005, 0.01 as learning rates. The model performance was evaluated based on MSE and maximum error.

**Performance evaluation.** The prediction capability of Model1 was evaluated based on a comparison with predictions from human experts. Specifically, here, Model1.2D was evaluated as the 2D colonies were easier to present to the experts. To this end, we implemented an application in Jupyter Notebooks that allowed convenient labeling for human experts (Fig 3). Similar to Model1.2D, the experts predicted the dispersion parameter on 2D colonies from data set A.

Five human experts—the authors of this paper—got image representations of 2D colonies. Five randomly selected batches of 100 2D colonies were predicted per expert. The task was to predict the dispersion parameter using a slider with steps of 0.05. The data was saved and compared between experts as well as with the predictions of Model1.2D. As the experts only

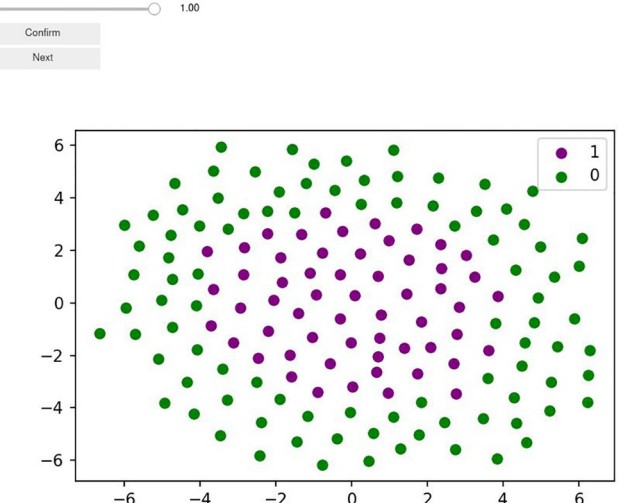

**Fig 3. Human expert pattern recognition.** Screenshot of the web-app created using Python and Jupyter Notebooks to enable human expert predictions of the dispersion parameter from the visualized cell coordinates and fates.

predicted using a slider with steps of 0.05 and the model predicted a continuous variable, the predictions needed to be made comparable. To this end, all model predictions were rounded to the next fitting step.

**Application to experimental data.** Model1.3D with the optimal hyperparameters contains two GSCConv layers with 450 and 150 nodes and relu and sigmoid activation functions, respectively. These are followed by a GlobalAttnSumPool layer. After that, two densely connected layers with 50 and 1 node(s) and "relu" and "sigmoid" activation, respectively, follow. In between these, a keras flattening layer is used to get one-dimensional output. Model1 was trained on data set B (Table 1) and applied to data set E to predict a dispersion parameter for *in vitro* organoids.

## Neural network for pattern reconstruction

For pattern reconstruction through spatial imputation of the fate of one cell, we implemented multilayer perceptrons with three hidden layers. The input data consists of a previously determined number of neighbors and their distance to the given cell. As output, we considered either one single neuron (Model2, Fig 4) or four neurons (Model3, Fig 4).

**Data conversion to input structure.** To transform data sets C, D and E (Table 1) to the required input format, a pipeline to preprocess the raw data was implemented. Based on the given coordinates of the cells, a variable number of nearest neighbors and their distance to the given cell were calculated. This input to the models contained the features cell ID, cell fate, and the calculated number of neighbors with their associated distance. The categorical labels of the experimental ICM organoids were transformed to numeric values (N+G+ = 0, N−G+ = 1, N−G− = 2, N+G− = 3), for better machine readability. Hence, the cell fate feature was binary for simulated data or quaternary for experimental data. The cell fate feature was used as the label in all models. The ID feature was dropped. The number of the remaining features was determined depending on the number of neighbors used as an input of the models. The data was randomly split in 80% training data and 20% test data.

**Model architecture.** Two different machine learning models were implemented and trained for pattern reconstruction. Model2 was trained with simulated data (data sets C and D,

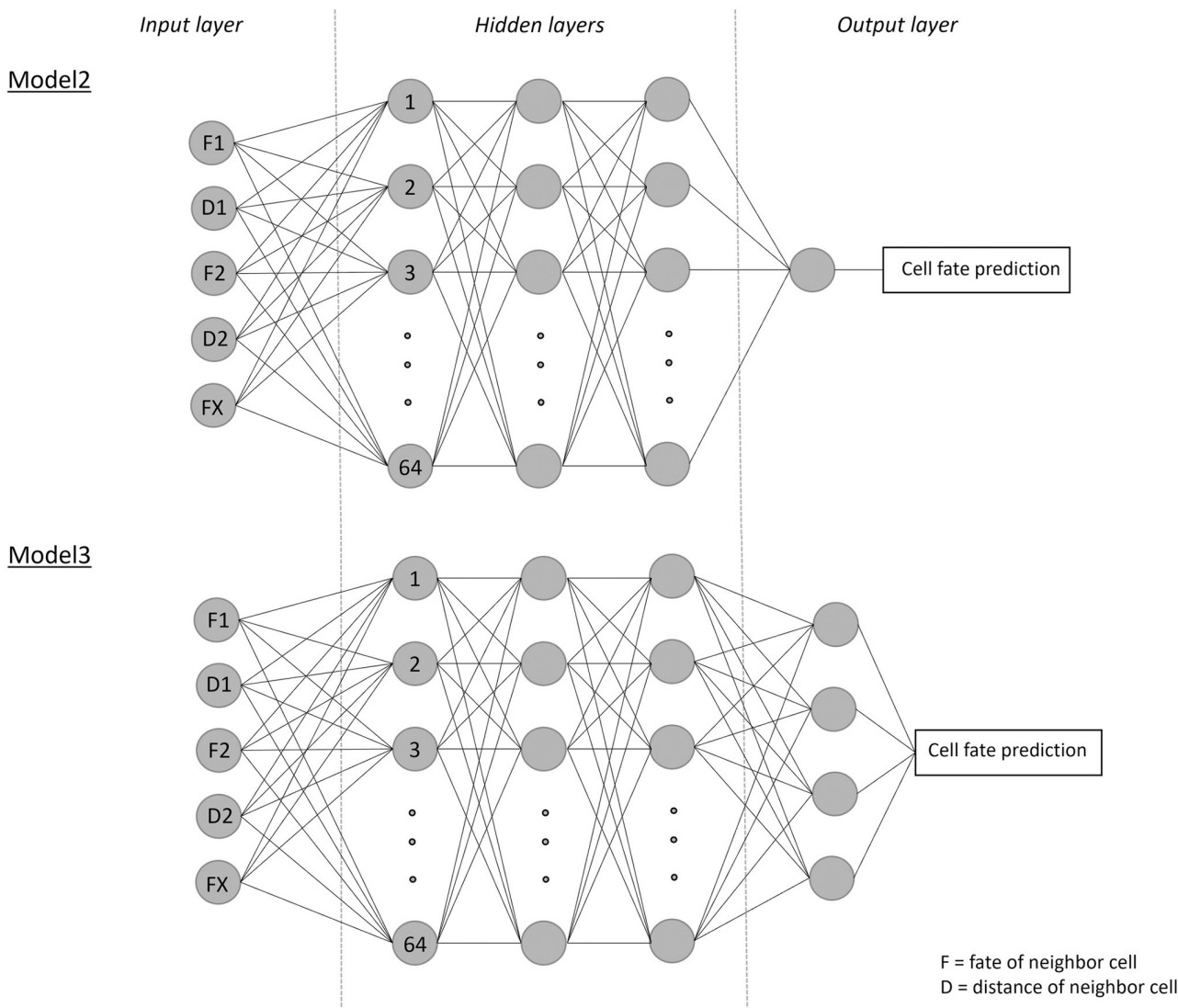

**Fig 4. General architecture of Model2 and Model3.** The cell fate (F1, F2, . . .) and the distance (D1, D2, . . .) of the nearest neighbor cells were taken as input features. The number of nodes in the input layer of both models was dependent on the number of neighbors taken as an input. Thereby, FX is representative for the additional cell fates and distances that we accounted for.

Table 1) and Model3 was trained with experimental data (data set E). Model2 consists of three hidden layers with 64 fully connected nodes each. As activation function, we chose the rectified linear unit (ReLU) function. Furthermore, we utilized the Adam optimizer [19] with a learning rate of 0.001. As a loss function, Tensorflows SparseCategoricalCrossentropy [20], which computes the cross entropy between the labels and predictions, was used. The output layer consists of two nodes, representing the label possibilities. The node with the higher value was taken as the prediction of the cell fate (Fig 4). Unlike Model1 from the Pattern Recognition part, which solves a regression problem and therefore uses MSE as a metric, this is a classification problem. Hence, we used the accuracy as a metric for Model2 and Model3. Model2 was trained with an autostop function, which monitored the validation loss. After 75 epochs without improvement of the validation loss, the training was stopped automatically. For easier understanding,

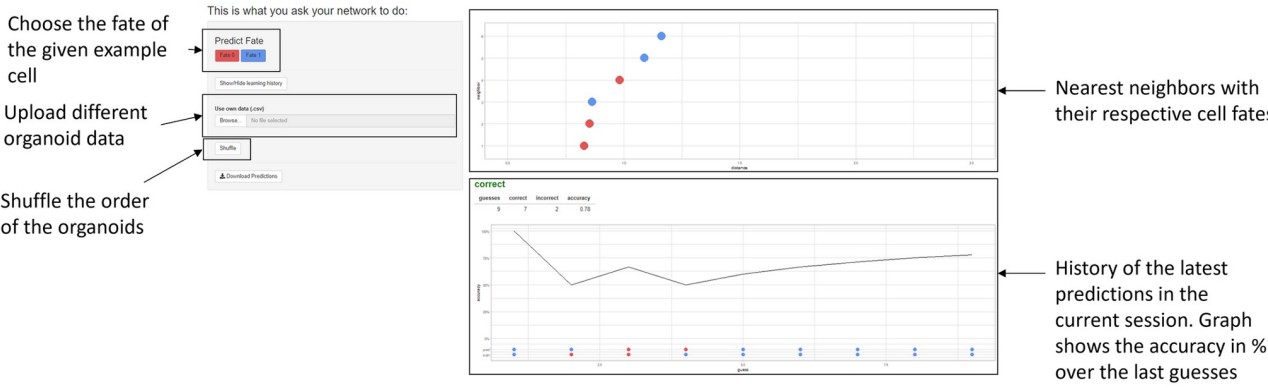

**Fig 5. Human expert pattern reconstruction through imputation.** Screenshot of the web-app created with shiny [22] to enable the human predictions of the fate of a given cell in a way that is comparable to the model.

Model2 will be referred to as Model2.2D when applied to simulated 2D data (data set C, Table 1) and Model2.3D when applied to simulated 3D data (data set D, Table 1).

Model3 was constructed similarly to Model2. It also uses three hidden layers with 64 fully connected nodes each. Again, the ReLU activation function and the Adam optimizer with a learning rate set to 0.001 were chosen. As a loss function, Tensorflows SparseCategoricalCrossentropy is used. In the output layer of Model3, four nodes represent the four label possibilities (Fig 4). The node with the largest value is taken as the prediction. Model3 was also trained with an autostop function, which monitored the validation loss. After 75 epochs without improvement of the validation loss, the training was stopped automatically. The best checkpoints were restored and used for testing the models.

**Performance evaluation.** The prediction capability of Model2 was evaluated based on a comparison with predictions from human experts. To this end, an application was developed in R [21]. Within the app, it is possible to display the data in the way it is passed to our model. The preprocessed 2D colony data for Model2 was taken as input in form of a csv file. After shuffling the order of the data, the cell fate could be predicted, based on the next neighbors displayed in a graph (Fig 5). The application was used to determine the human accuracy in predicting the different patterns. The seven nearest neighbors were displayed. For every dispersion parameter $q$, 100 predictions were made by five human experts—the authors of this paper.

## Results

### Spatial statistics

During the preimplantation phase of mouse embryogenesis, two sequential rounds of cell fate decisions occur that result in three cell populations (reviewed in [23]). First, cells become either trophectoderm or inner cell mass (ICM) cells. In the subsequent blastocyst stage, ICM cells segregate into either primitive endoderm or epiblast cells. Descendants of trophectoderm cells form the fetal part of the placenta, while primitive endoderm cells are mainly precursors of the endodermal cells of the yolk sac. Epiblast cells mainly give rise to the embryo proper.

Mouse ICM organoids are three-dimensional aggregates of mouse embryonic stem cells from the epiblast portion of the ICM (Fig 6) [5]. These cells have been engineered to

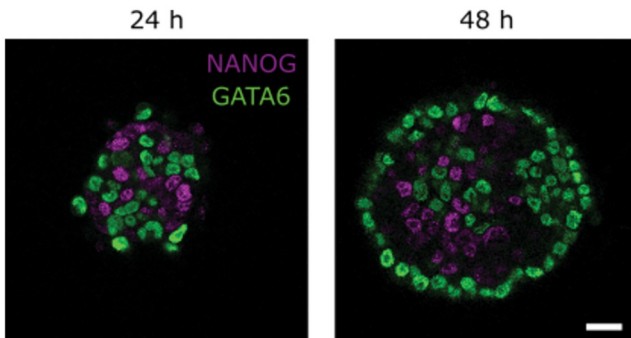

**Fig 6. Mouse ICM organoids 24 h and 48 h past formation.** NANOG (magenta) and GATA6 (green) are mutually exclusively expressed in three-dimensional aggregates of mouse embryonic stem cells. Images show a single slice from the aggregate's center. Microscope: Zeiss LSM780; objective: 63x/1.40 oil; scale bar: 20 μm.

replicate the cell differentiation towards epiblast and primitive endoderm cells [3]. In our previous work, we have identified local clustering of epiblast and primitive endoderm like cells in the ICM organoids which is comparable to the cell fate distributions in the mouse embryo.

Moran's index provides a measure for spatial autocorrelation for patterns of two cell types [16]. We find that 24 h after organoid formation, the Moran's index is consistently larger than 0, indicating a non-random distribution of the cells (Fig 7, data set E). For 48 h organoids, the average Moran's index increases further, indicating stronger clustering of the two cell types.

To focus on the radial distribution of the two cell types, we plotted pair correlation functions (PCF) for all organoids (S1 and S2 Figs, https://schardts.github.io/Organoids24h and https://schardts.github.io/Organoids48h). For 24 h, we get a mixture with some PCFs that are close to one, indicating a random distribution while others show signs of radial separation. For 48 h organoids, the results are clearer. The PCFs for 34 out of 42 organoids show a pattern with a small drop at the beginning and a sharp rise at the maximum pair distances for GATA6 expressing cells, indicating a preferential distribution of these cells at the organoid rim. The PCFs for NANOG expressing cells in these organoids show a monotonous decrease, indicating a depletion of these cells along the boundary of the organoids.

To investigate how these patterns relate to patterns generated by our cell differentiation model, we performed simulations on the geometry of the experimental data with the appropriate cell fate ratio. We chose six different 48 h organoids. Three of these organoids could be

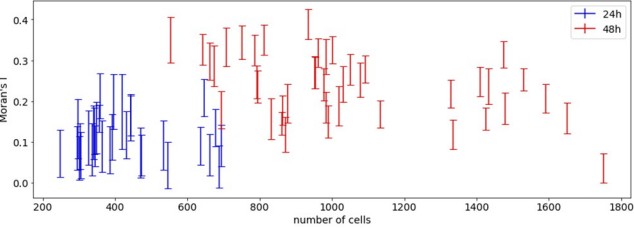

**Fig 7. Range of Moran's index for each ICM organoid with respect to cell number.** Each bar shows the resulting range from minimum to maximum of Moran's index for 1000 samples of randomly chosen fates for N−G− and N+G+ cells in data set E (see materials and methods for more details). (Figure from [33]).

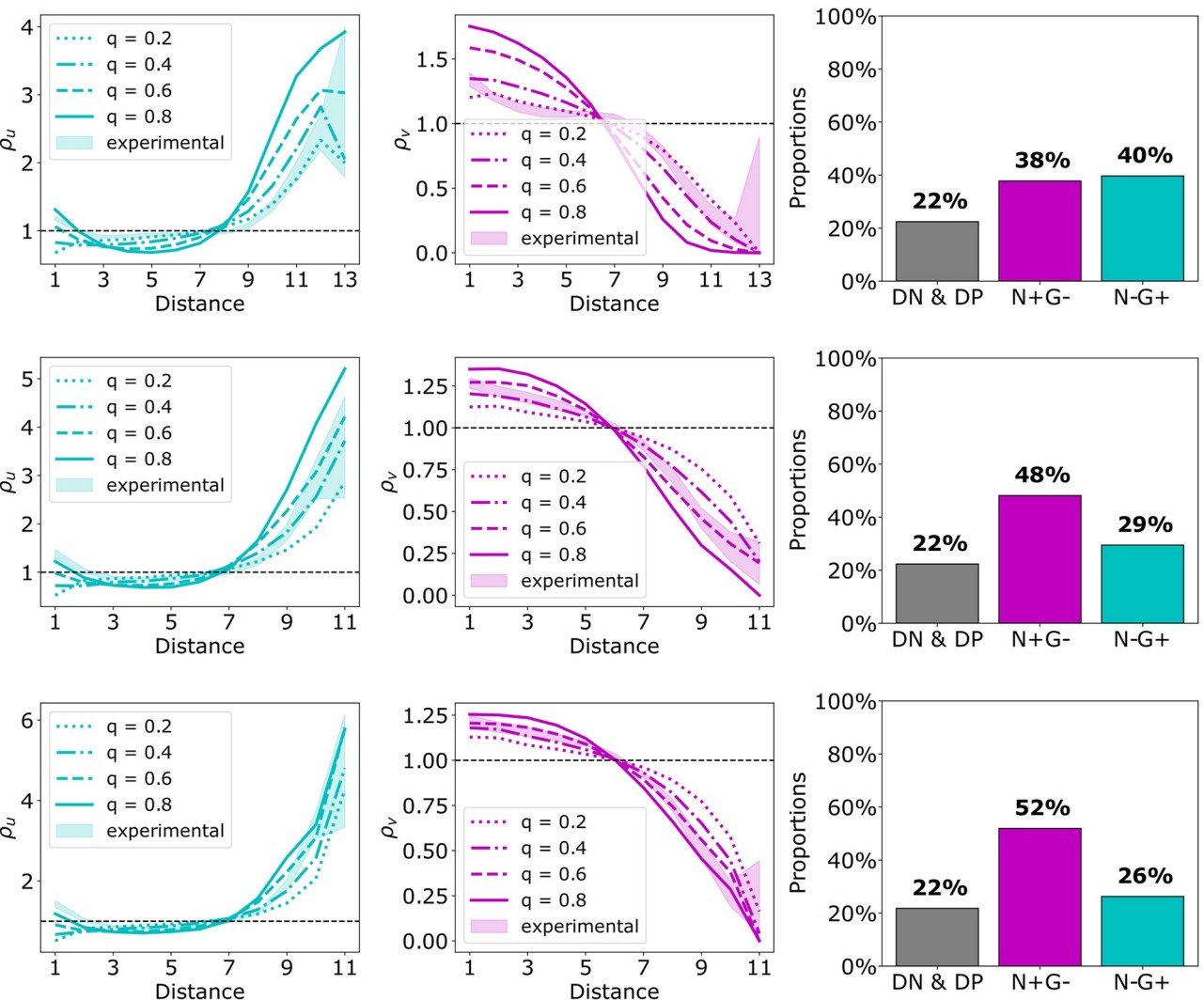

**Fig 8. Three examples of a good match between experimental data and simulations.** In each row, we observe the data of a single 48 h ICM organoid from [5]. From left to right, the PCFs $\rho_u$ and $\rho_v$ and the cell type proportions are visualized. The PCF plots include PCFs for different dispersion parameters $q$ as well as the PCF envelope generated by the maximum and minimum for 1000 different samples of randomly choosing cell fates for N−G− and N+G+ cells (see materials and methods for more details). Organoid IDs corresponding to S2 Fig from top to bottom are 2, 32 and 36. (Figure from [33]).

readily matched by the simulations (Fig 8), while the PCFs for the other three were hard to reproduce by the simulations (Fig 9).

Next, we applied Moran's index to investigate the matching of the experimental data and the simulations for the same six organoids (Fig 10). For the simulations for $q >= 0.4$, we see that Moran's index depends on the organoid and hence its cell fate ratio and geometry. Based on the Moran's index, we find good matches between experimental data and simulation results for five out of the six organoids. We then focused on the simulations for $q$ values predicted based on the PCFs for the good fitting organoids. We see that these Moran's indices lie below those of the experimental data. Organoid 36 shows the best match. Hence, predicting the $q$ value based on the Moran's indices for these organoids would provide a larger $q$ than predictions based on the PCFs.

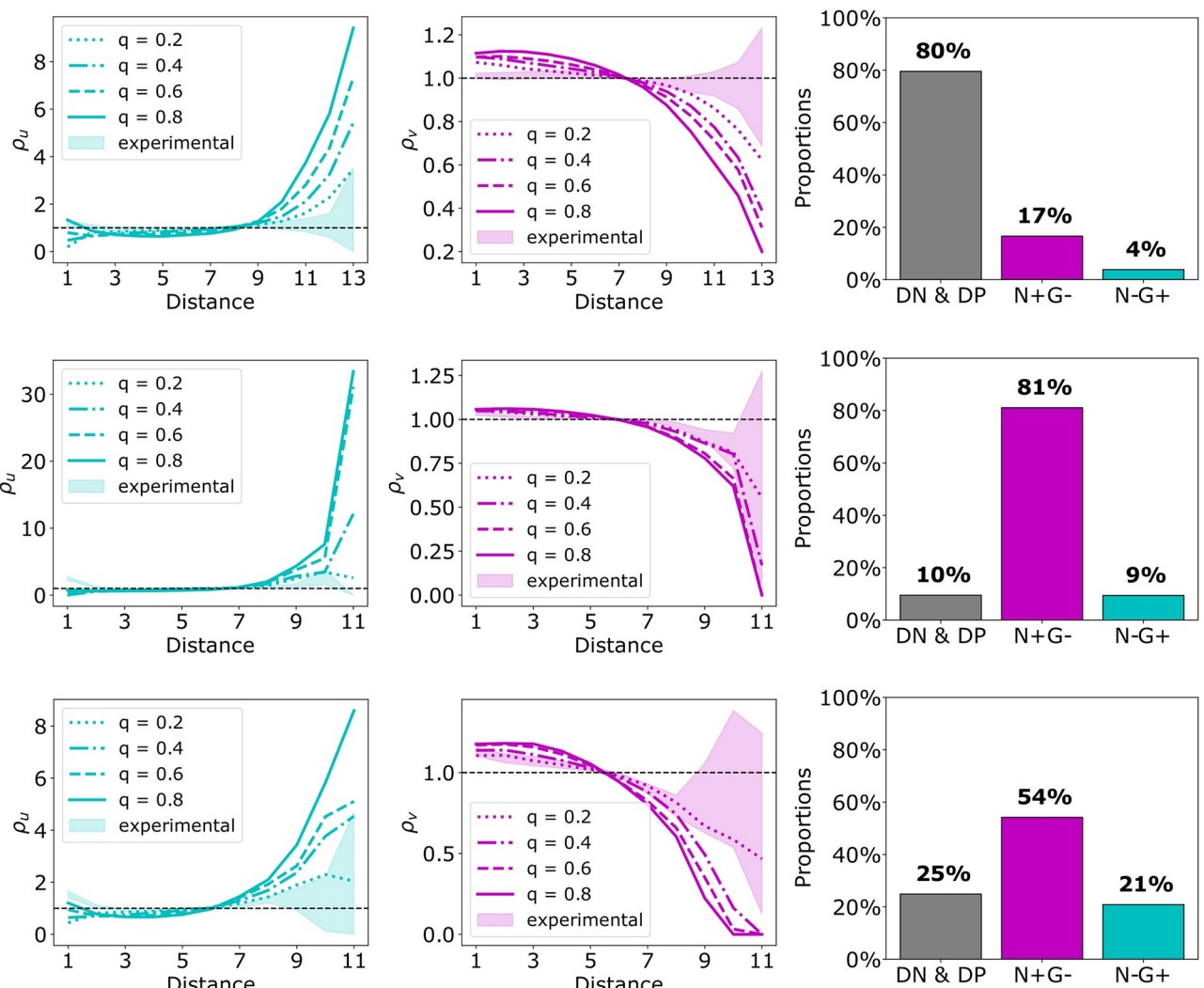

**Fig 9. Three examples for which the match between experimental data and simulations is challenging.** In each row, we observe the data of a single 48 h ICM organoid from [5]. From left to right, the PCFs $\rho_u$ and $\rho_v$ and the cell type proportions are visualized. The PCF plots include PCFs for different dispersion parameters $q$ as well as the PCF envelope generated by minimum and maximum of 1000 different samples of randomly choosing cell fates for N−G− and N+G+ cells (see materials and methods for more details). Organoid IDs corresponding to S2 Fig from top to bottom are 1, 12 and 39. (Figure from [33]).

In summary, both Moran's index and PCF indicate that some 24 h and most 48 h organoids show some kind of sorted state. Comparing the experimental data with results for simulations produces ambiguous results.

## Pattern recognition

Pair correlation functions and Moran's index provide summary statistics of the cell fate patterns. To enable direct linking of the spatial patterns to the dispersion parameter $q$ of our cell differentiation model, we constructed a graph neural network for characterization of binary spatial patterns (Fig 11). The input data contains positional information and a binary fate for every cell and was converted into graphs. The output consists of the single pattern characterization parameter. The neural network was constructed using TensorFlow and spektral. For

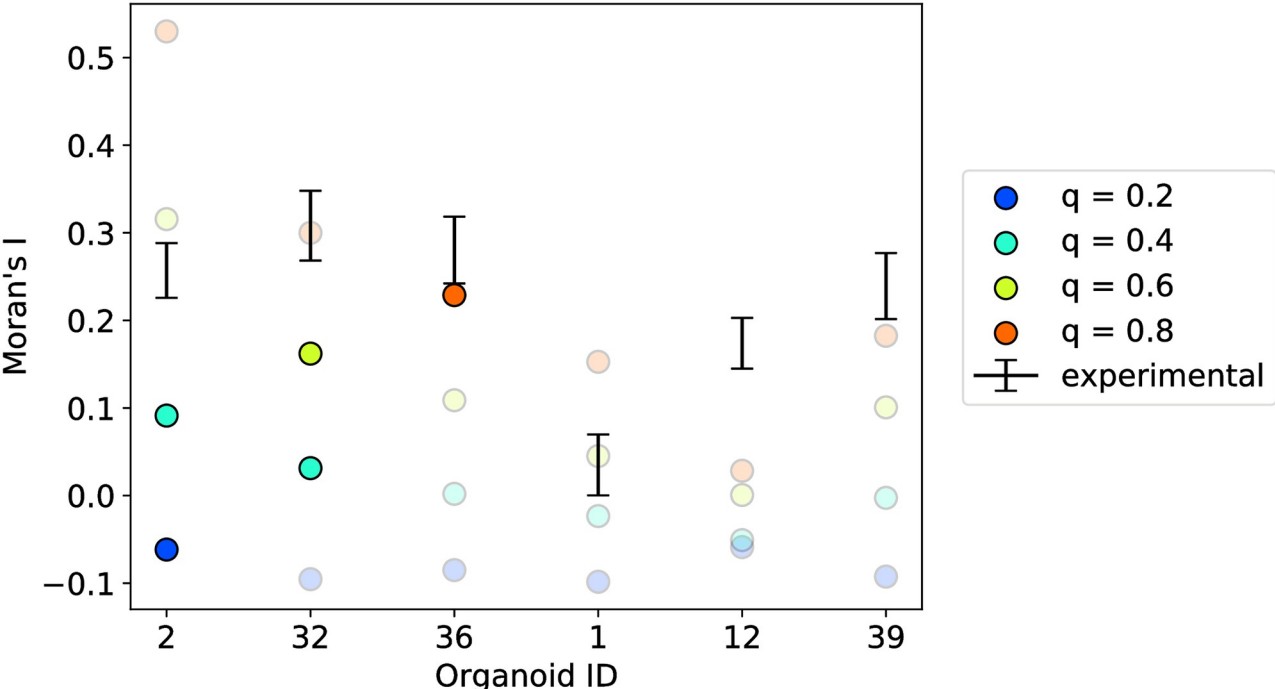

**Fig 10. Matching experimental data and simulations based on Moran's index and PCFs.** The three examples on the left represent the positive examples, where the PCF looked promising, whereas the three on the right are the ones that were more challenging. Colored disks indicate Moran's indices for simulations results for the respective organoid geometry and *q* value. Markers with higher opacity highlight the Moran's indices for simulations with values of *q* that were predicted based on the PCFs. The range for the experimental data was generated by minimum and maximum of 1000 different samples of randomly choosing cell fates for N−G− and N+G+ cells (see materials and methods for more details). Organoid IDs corresponding to S2 Fig. (Figure from [33]).

training and testing, we used synthetic data based on our cell differentiation model. We performed simulations for the model for different dispersion parameters *q*, resulting in binary cell fate distributions ranging from checkerboard-like to engulfing. To train our GNN, we used the spatial distributions as input and the value of *q* as output.

To optimize model performance, different hyperparameters were systematically tested. The hyperparameter tests revealed that the most accurate model employed two spektral GSCConv layers with 450 and 150 channels and a "relu" and "sigmoid" activation function,

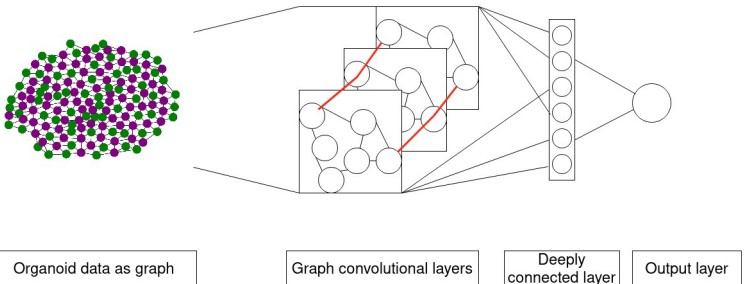

**Fig 11. Schematic overview of the pattern recognition approach.** The data is passed as a graph containing the cells as nodes and their cell fates as node features. Neighboring cells are connected by an edge. A combination of graph convolutional layers and densely connected layers generates a prediction of the dispersion parameter *q* as output.

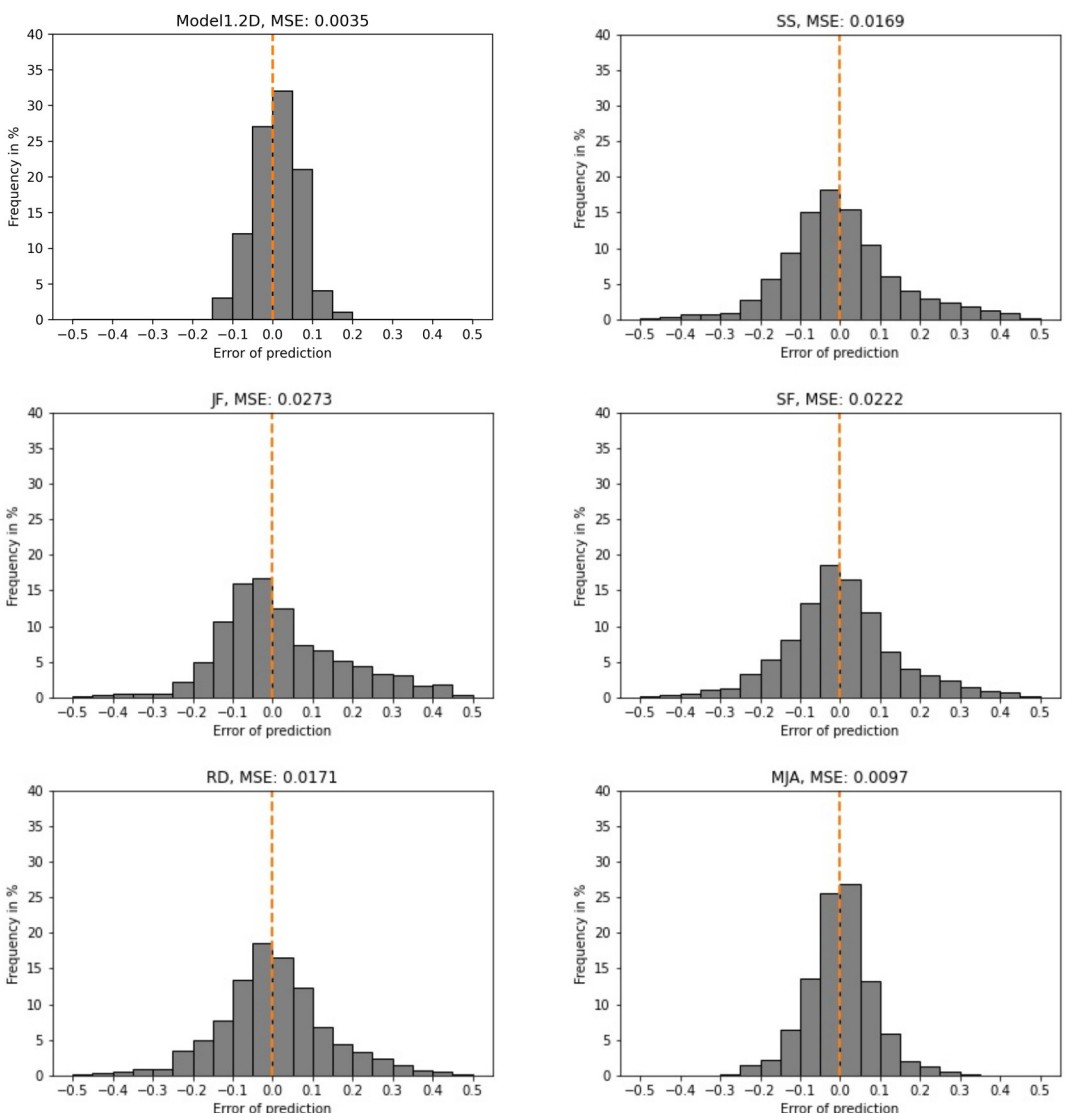

**Fig 12. Model1.2D performs better than all human experts.** Histograms of error as difference between prediction and true value, and MSE (plot label) for the GNN (top left) and all experts for simulated 2D colonies from data set A (Table 1). The dashed orange line indicates zero error.

respectively. Following, one spektral GlobalAttnSumPool layer was used for pooling, followed by two dense layers, with 150 and 50 nodes and "relu" and "sigmoid" activation functions, respectively.

Model1.2D employed this architecture and was trained and tested on data from 2D simulated colonies (data set A, Table 1). It achieved a mean squared error (MSE) of 0.0035 on testing data, which was not used in training (Fig 12).

To assess how predictable the dispersion parameter is and how difficult this problem is for humans, five experts, the authors of this paper, predicted the dispersion parameters for 500 2D colonies from data set A (Table 1). To asses the performance of Model1.2D, the expert predictions were compared with those of the model in terms of MSE (Fig 12). The best performance of an expert lead to an MSE of 0.0097, while Model1.2D showed an MSE of 0.0035. Hence,

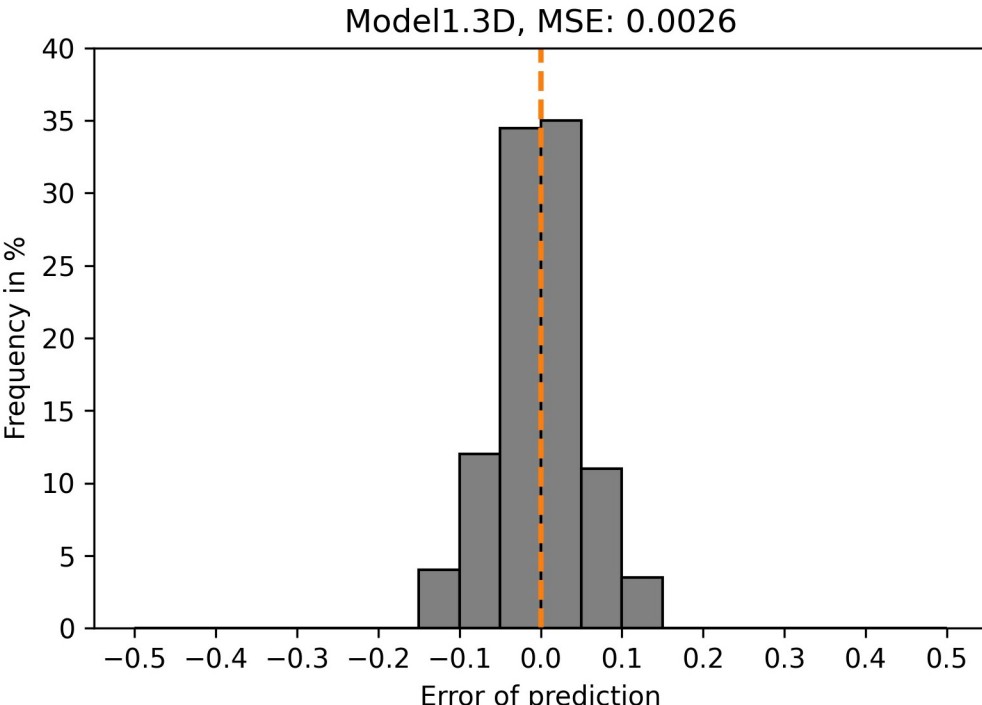

**Fig 13. Model1.3D exhibits a low MSE.** Histogram of error as difference between prediction and true value, and MSE (plot label) for the GNN trained and tested on data set B (Table 1). The dashed orange line indicates zero error.

Model1.2D generally predicts more closely to the truth than all humans. Interestingly, for humans and the GNN, predictions for lower values of $q$ were more disperse than for larger $q$ (S3 Fig). This indicates that patterns for lower $q$ are harder to identify.

For more accurate resemblance of *in vitro* organoids, Model1 was also trained and tested on three-dimensional simulated organoids of variable sizes (data set B, Table 1). The evaluation of this Model1.3D revealed an MSE of 0.0026 (Fig 13). Overall, the differences between the true and the predicted dispersion parameters are very small when looking at Model1.3D. Over 70% of predictions are wrong by less than 0.05. The maximum error is smaller than 0.15.

To assess the applicability of Model1.3D to experimental data, it was used to predict a dispersion parameter for our data from ICM organoids that were cultured for 24 h or 48 h (data set E, Table 1, [5]). Generally, all *in vitro* organoids are predicted to have a low dispersion parameter in the range of 0 to 0.5 (Fig 14).

The maximum predicted dispersion parameter for 24 h organoids was less than 0.4. Furthermore, 24 h organoids showed peaks in very low and mid-range predictions. A large part of 48 h organoids was predicted to have dispersion parameters in the range of 0 to 0.2, with the frequency declining for a larger dispersion value. The distribution was slightly shifted towards a higher predicted dispersion parameter compared to 24 h organoids. Around 12% of 48 h organoids were predicted to have a dispersion parameter in the range of 0.4 to 0.5.

We successfully constructed a GNN and trained it on simulated 2D colonies and organoids to predict a signal dispersion parameter that was in place during simulation. The model performed better than five human experts and showed a low MSE. The model that was trained on simulated data was applied to predict dispersion parameters for *in vitro* ICM organoids. The resulting dispersion parameters were in the range between 0 and 0.5 both for 24 h and 48 h

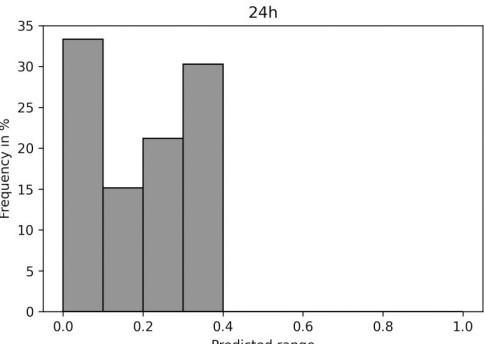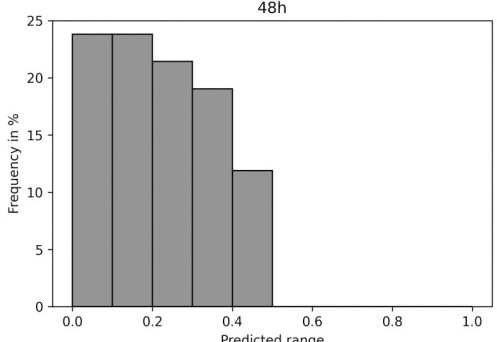

**Fig 14. *in vitro* organoids are predicted to have rather low dispersion parameter.** Dispersion parameter values predicted by Model 1.3D for 24 h (left) and 48 h (right) organoids.

organoids. This suggests that in ICM organoids, the local cell neighborhood is linked to the fate of a given cell and differs from our results from the summary statistics that suggest interactions at different ranges.

If a cell's fate is linked to the local cell neighborhood, it should be possible to predict it's fate based on the information from the neighbors. To test this hypothesis, we next implemented a model for pattern reconstruction through spatial imputation. The task of the model was to predict the fate of a given cell based on the fates of neighboring cells.

## Pattern reconstruction

For pattern reconstruction through spatial imputation, three different machine learning models were constructed. Each of these three models was built according to the same general scheme (Fig 15). The models were designed to predict the fate of a given cell. A set of neighboring cells served as input with their fates and their corresponding distances to the cell in question. These machine learning models were trained and tested on simulated 2D colony and organoid data (data set C, D, Table 1) as well as experimental ICM organoid data (data set E, Table 1).

The initial goal in pattern reconstruction concerned the fate of an individual cell and how it relates to its environment. Specifically, we wanted to understand how the number of nearest

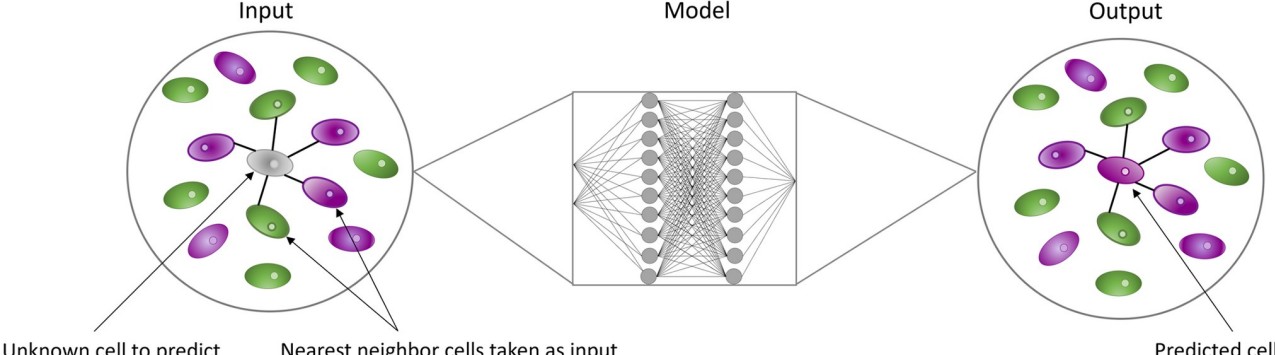

**Fig 15. Schematic overview of the pattern reconstruction approach.** The color of the cells indicates their cell fates. Highlighted cells are used as input for the machine learning algorithm which outputs a prediction for the grey cell.

cells considered in our model impacts the result of the predictions. For this purpose, we first focused on the data points for $q = 0.1$ from the simulated 2D data (data set C). Model2.2D was used to test how many neighbors the network needs, to make an accurate prediction. Therefore, different numbers of neighbors were taken as input. Since including the coordinates of the cells as cartesian or radial decreased the accuracy of the model predictions, the spatial distribution of the neighbors was not considered. From one to six neighbors, there was a significant increase in accuracy. The best testing accuracy with Model2.2D was achieved for seven neighbors (Fig 16). Adding more neighbors showed no improvement of the testing accuracy. The typical value for cells in contact with a given cell in this data set is six.

Since there is not only one expression pattern of cells in experimental data, it was necessary to investigate how well Model2.2D performs with different simulated patterns. Motivated by the findings of the first experiment, seven nearest neighbors were used as input. The model was then trained and tested on simulated 2D colony data with different $q$. Model 2.2D performed best on data with low $q$'s up to $q = 0.3$ and the highest $q$ with $q = 0.9$ (Fig 17). The patterns in the higher range of $q$ ($q \in \{0.6, 0.7, 0.8\}$) resulted in a bad performance of Model2.2D, with a testing accuracy below 80%. The predictability of the simulated 2D colony data for $q = 0.4$ and $q = 0.5$ showed a mid ($< 90\%$) to low ($< 85\%$) testing accuracy.

The previous experiment showed that our model is capable of working with 2D data. Since our microscopy data is three-dimensional, new data were simulated in 3D (data set D). With the aim of determining the influence of the neighboring cells on the predictions, these data were examined according to the same scheme as the 2D data. Due to the general architecture of Model2, it could also be used with the larger 3D data volumes. It will be referred to as Model2.3D when handling 3D data.

To test the influence of the number of included neighbors on the performance of the model, the data points for $q = 0.1$ from the simulated 3D organoid data (data set D) were used. The best results were obtained with 12 to 14 neighbors included. Here, the validation

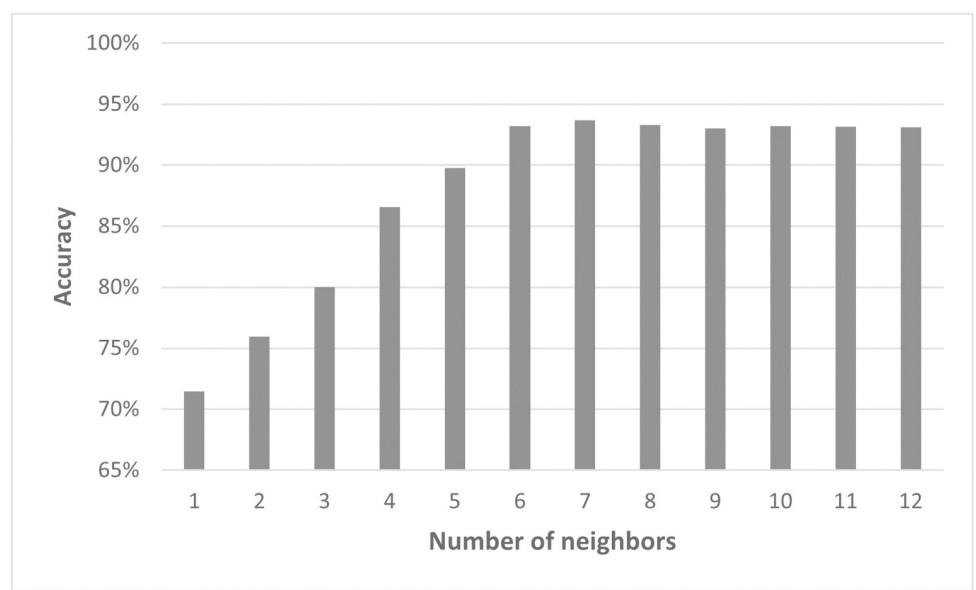

**Fig 16. Model2.2D performs best for seven neighbors.** Overview of the testing accuracy of Model2.2D predicting simulated 2D colonies with q=0.1 dependent on the number of neighbors taken as an input for the model. Please note that for a clearer display, the y-axis starts at 65%.

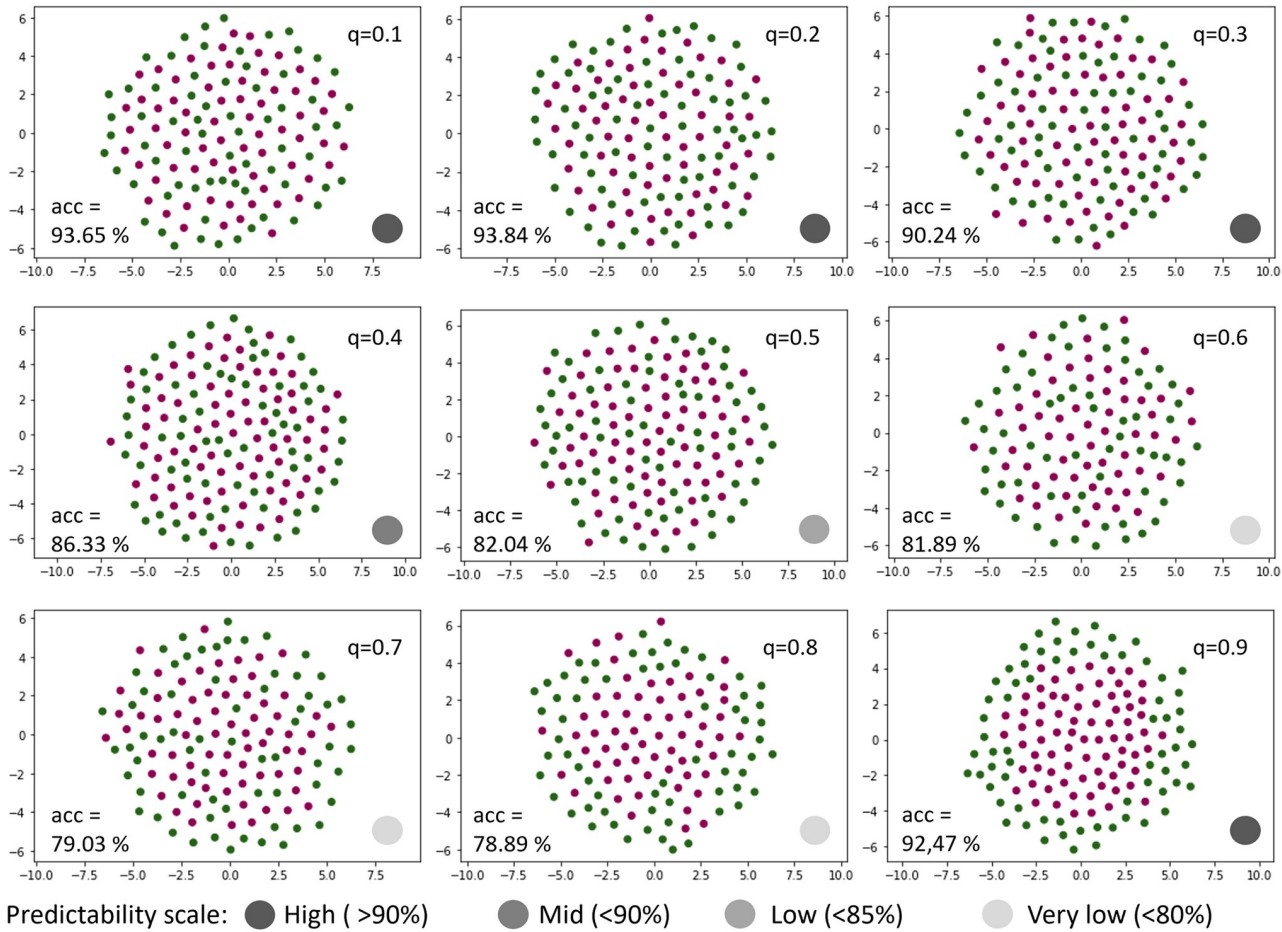

**Fig 17. Low and high values of _q_ show best predictability with Model2.2D.** Overview of the predictability of simulated 2D colonies with different dispersion parameters _q_. All trainings and predictions were made under the same conditions with Model2.2D. The greyscale color gradient indicates the predictability of the different data, based on the accuracy.

accuracy was over 90%. If fewer than eleven neighbors were used, the accuracy fell to 86% or lower (Fig 18). Using more than 14 neighbors, on the other hand, showed no improvement in performance. The typical value of cells in contact with a given cell in this data set is 14. Extending the analysis to all values of _q_ showed a similar trend (S4 Fig). Interestingly, the accuracy for lower _q_ was more sensitive to the number of neighbors than values of _q_ greater than 0.5.

It could be shown that our models work with the simulated data, but comparative evaluation of the performance is missing. Since there are no analogous approaches to our knowledge, the results were compared with human predictions from five experts. The accuracy of human and Model2.2D applied to data set C (Table 1) showed a similar trend (Figs 19 and S5, Model2.2D and Human). The accuracy of the model was almost always 15% higher than that of the human. The highest average accuracy of 90% reached by the human experts was obtained on 2D colony data with _q_ = 0.9. The model trained with 3D organoid data showed a different trend in performance. Here, the worst accuracy was measured for the organoids between _q_ = 0.3 and _q_ = 0.6. For the remaining organoid data sets, the model achieved an accuracy of at least 90%. (Fig 19, Model 2.3D).

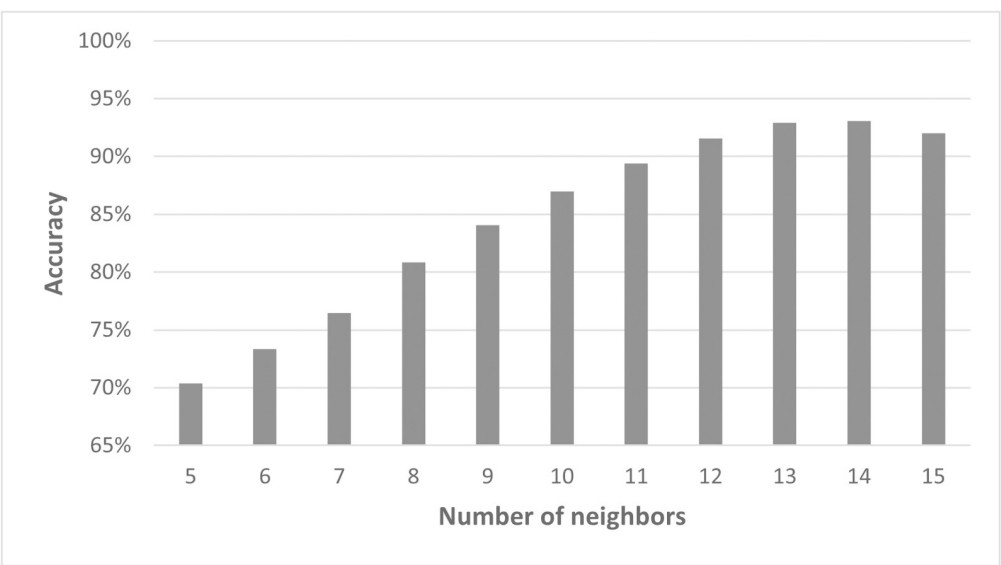

**Fig 18. Model2.3D for simulated 3D organoids performs best for 12–15 neighbors.** Overview of the accuracy of Model2.3D predicting simulated 3D organoids with $q$=0.1 dependent on the number of neighbors taken as an input for the model. Please note that for a clearer display, the y-axis starts at 65%.

While in the simulated data, the different patterns were split into the individual $q's$, in experimental data these can occur all at once. Therefore, it was tested whether Model2.3D is still able to predict the cell fate of a single cell when trained with all simulated data D (Table 1). The results show the highest accuracy for values of $q > 0.5$ (Fig 20). This is in good agreement with our pattern recognition findings that showed that patterns for lower $q$ are more difficult to recognize. To get a better understanding of this result, we performed a detailed cross-training analysis. We split the training data according to the values of $q$ and tested how well such a model predicts the cell fates in test data sets for the different $q$. The results revealed that the training data sets for lower values of $q$ are very specific (S6 Fig). For $q$ between 0.6 and 0.9, the training data contains enough information to achieve an accuracy of at least 89% for all $q$ in that range.

The models proved their ability to reconstruct binary cell differentiation patterns in simulated 2D and 3D data with high accuracy. Furthermore, their capabilities exceeded those of human experts. Now, we needed to examine the experimental data with a similar model. So far, we have simplified the cell populations in ICM organoids into two classes. However, in the field of mouse embryogenesis, it is common practice to separate the cells into four categories. Hence, we obtain epiblast precursors (N+G−), characterized by high levels of the transcription factor NANOG and low levels of GATA6. Likewise, primitive endoderm precursors (N−G+) are characterized by high levels of GATA6 and low levels of NANOG. Additionally, there are double positive (N+G+) and double negative cells (N−G−) which express high and low levels of both transcription factors, respectively. Thus, we adapted our model such that it could output four cell fates instead of two.

Model3 was then first used to examine data set E (Table 1) using the same scheme as previously used for the simulated data. The results showed that at least the four nearest neighbor cells must be taken as input to achieve an accuracy of over 71%. The best result was achieved with nine neighbors included. Increasing the number of neighbors above ten did not improve the accuracy of the model. The lowest accuracy was achieved using only the single nearest

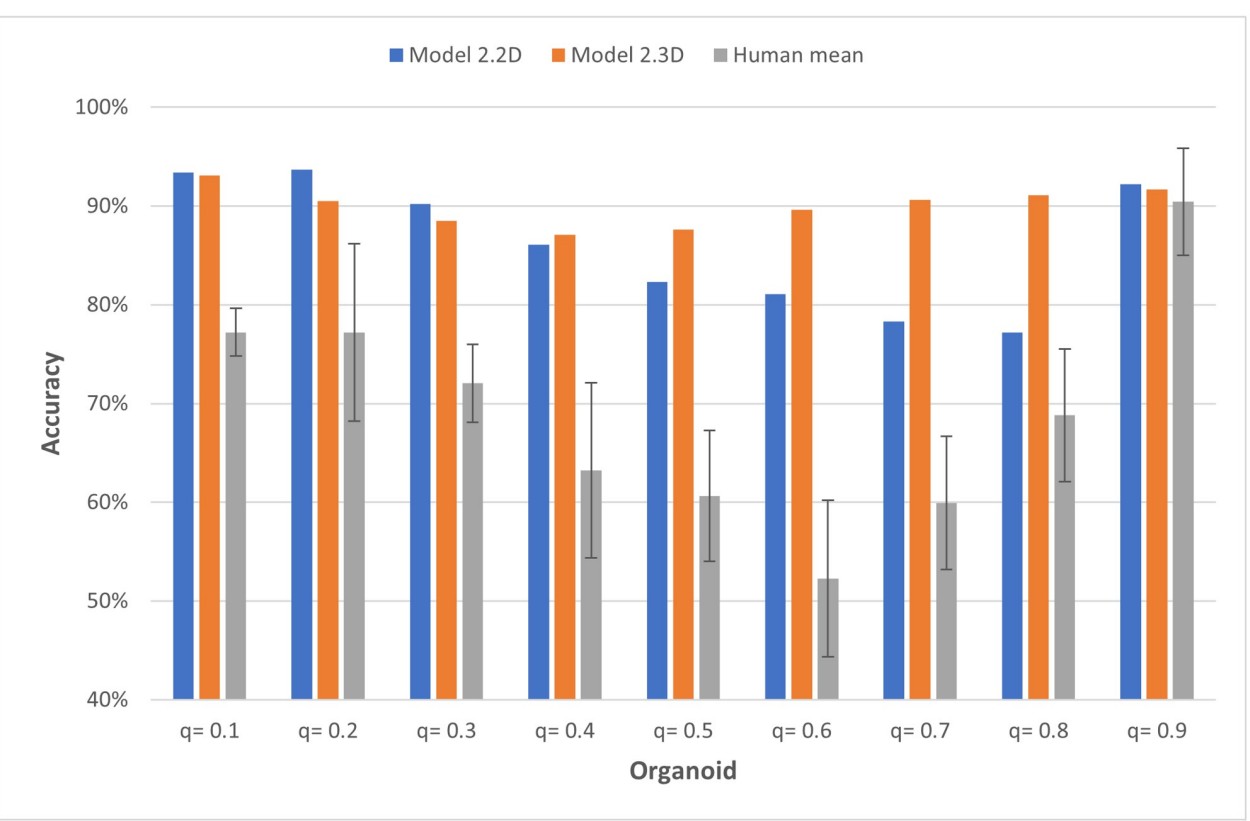

**Fig 19. Model 2.2D and 2.3D outperform human expert predictions.** Accuracy of Model2.2 on different simulated 2D colony and organoid data in comparison with humans. Both, Model2.2D and Human were tested with seven neighbors, using two-dimensional simulated colony data (data set C, Table 1). Model2.3D was run with 14 neighbors on three-dimensional simulated organoid data (data set D, Table 1). Please note that for a clearer display, the y-axis starts at 40%.

neighbor. This was around 66% (Fig 21). The typical value of cells in contact with a given cell in this data set is 14.

In the next step, we tried to improve the performance of Model3 with experimental data via feature engineering. Separating data set E (Table 1) into 24 h and 48 h organoids and training each independently with Model3 did not improve the accuracy. Similarly, it made no difference whether the 24/48 h label was used as a feature when training the model.

To localize the errors of the model, six representative organoids were taken from the data set. Three 24 h organoids and three 48 h organoids were selected, each from a different batch. All organoids were predicted with the same fully trained Model3. In the predictions of the 24 h organoids, the errors occurred primarily in the interior of the organoids and in smaller clusters of cells with a different cell fate. The predictions of the 48 h organoids showed the errors especially in the interior of the organoids and in between the mantle formed by the primitive endoderm and the epiblast precursor cells. Interactive visualizations of the accurate and inaccurate predictions are available here: https://rodirk.github.io/patternReconstruction/.

After locating the errors in the organoids, confusion matrices were used to examine whether there were any noticeable patterns in the prediction of Model3. First, we considered the standard confusion matrix (Fig 22). It showed that for all labels, the correct prediction presented the majority of the predictions. However, looking at the top row, we find that in a large number of cases the label N+G+ is wrongly predicted for true labels N−G+ and N+G−. This

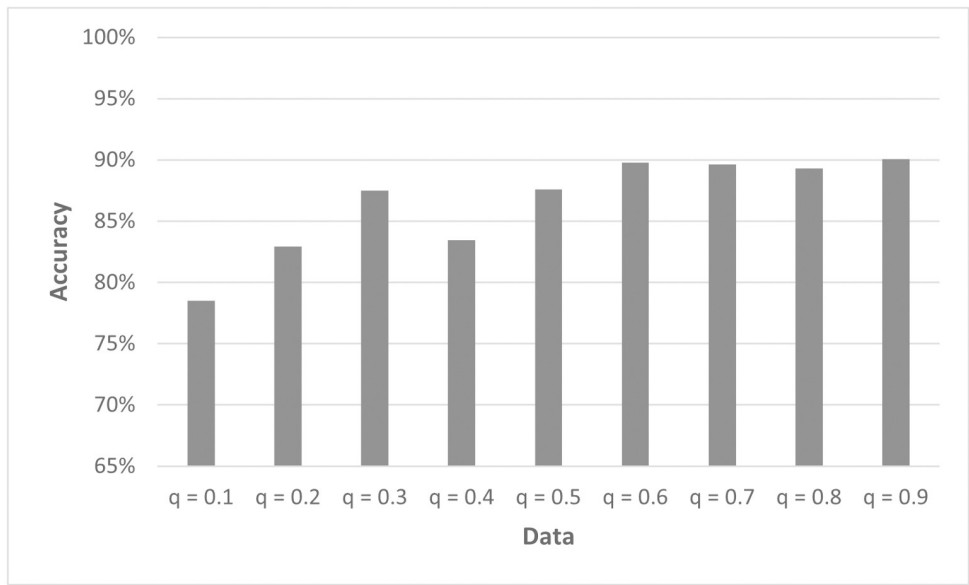

**Fig 20. Model 2.3D trained on data set D (Table 1) performs better for high *q*s.** Accuracy of Model2.3D for different values of *q*. As training data, we used the complete data set D (Table 1) without splitting it according to the value of *q*. Please note that for a clearer display, the y-axis starts at 65%.

becomes even clearer when we adapt the confusion matrix such that the entries in each row are normalized to 1. While the remaining labels could be predicted with at least 72% accuracy, the accuracy of Model3 was about 30% for the N+G+ label. Analyzing the organoids individually underlines this trend but also highlights the large variability between organoids (S7 Fig).

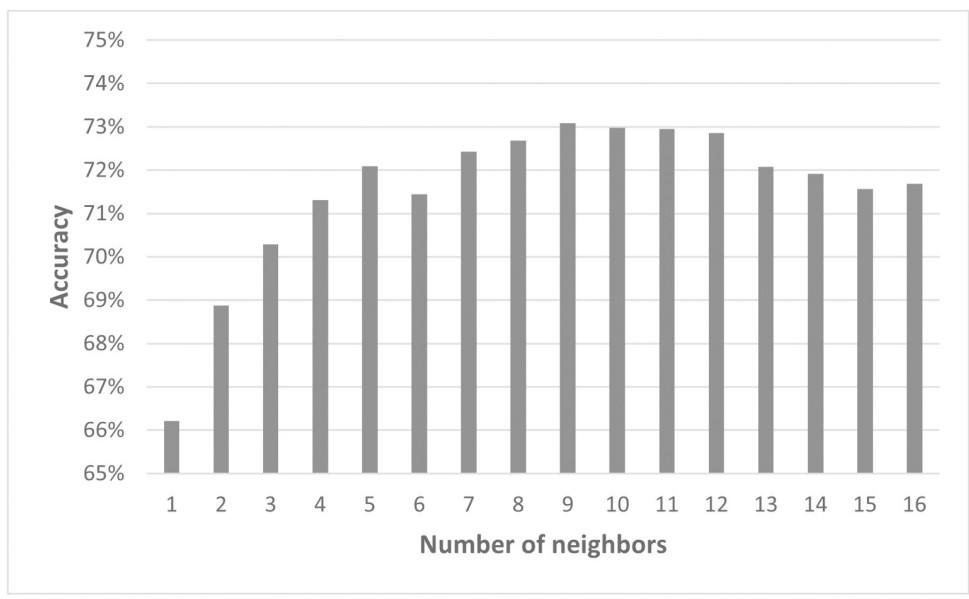

**Fig 21. Model3 for ICM organoids performs best for 8–12 neighbors.** Overview of the accuracy of Model3 predicting ICM organoid data dependent on the number of neighbors taken as an input for the model. Please note that for a clearer display, the y-axis starts at 65%.

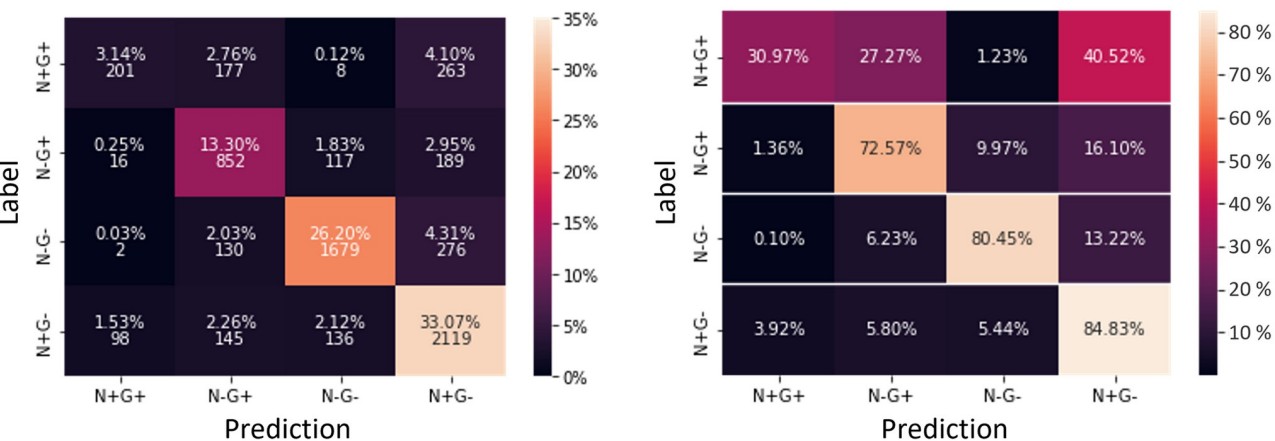

**Fig 22. Confusion Matrix of cells pooled from six ICM organoids for Model2.3D.** All predictions were plotted against all labels, shown as percentages and totals in the confusion matrix (left). Confusion matrix in which the row entries were normalized to 1 (right). Test predictions made with the fully trained Model3 on cells from six different ICM organoids.

We successfully constructed a multilayer perceptron to predict the fate of a given cell based on the fates of its neighbors. We trained and tested it on simulated data to find that we achieve the best performance if seven neighbors for 2D data and 13–14 neighbors for 3D data are considered. These numbers correspond to the typical number of cells in direct contact with a given cell in the simulated tissues. The model outperforms human predictions on 2D data. For the ICM organoid data, we obtained an accuracy of more than 70% for nine neighbors that decreases for increasing neighbor number.

## Discussion

We successfully developed deep learning models that can recognize patterns ranging from checkerboard-like to engulfing and can reconstruct patterns through spatial imputation of the fate of a cell based on information on close neighbors. Application of our models to ICM organoid data revealed local cell fate patterns that are consistent with a distance-based signaling mechanism with a short dispersion range. Our work highlights how the combination of mathematical modelling and machine learning enables linking cell fate patterns to potential underlying mechanisms.

### Pair correlation functions and Moran's index provide information on radial segregation and local clustering for *in vitro* and simulated patterns

The pair correlation functions provided a good visualisation of the radial segregation of the cell fate patterns in ICM organoids. While in the PCFs for 24 h organoids, there was rarely a clear pattern, the PCFs for 48 h showed a clear segregation in almost all organoids. The Moran's index provides a measure for spatial auto-correlation. It indicated clustering in the organoids that was more pronounced after 48 h than after 24 h. These findings are consistent with our previous observation that ICM organoids show local cell fate clustering and the visual impression that 48 h organoids show an engulfing pattern [5]. For the simulated patterns we observe an increase in spatial segregation with increasing dispersion parameter $q$ in agreement with [12]. The Moran's index for simulated data also increases with increasing $q$. However, patterns for the same $q$ but different organoid geometry result in different Moran's indices.

## A graph neural network accurately predicts the dispersion ranges for simulated patterns

A graph neural network could accurately predict the dispersion parameter for different patterns that were simulated for a distance-based signaling mechanism. It performed substantially better than five human experts in predicting the parameter. This shows that the recognition of patterns that are difficult to quantify by the human eye benefits from alternative quantification methods.

We first tried to use multilayer perceptron (MLP) models for the prediction of the dispersion parameter. MLPs have been successfully used for image classification [24], which is also a pattern recognition task. However, none of the MLPs that we tried for our pattern recognition task reached a satisfactory performance. The MLP used individual cell coordinates and fates as training and input data. Unsatisfactory results led to the conclusion that the information needed for pattern quantification was insufficient.

We then reasoned that intercellular relations in organoids can be well represented in a graph structure. Graph neural networks (GNN) have been developed for training on and classification of graphs [25]. They have been applied on numerous tasks, including prediction tasks in the context of social networks or prediction of molecule characteristics [26]. Here, we introduced graph neural networks for classification of patterns of whole organoids based on the interactions and fates of the constituent cells. Contrary to the MLP, the graph neural network provided satisfactory accuracy. The neighborhood relations and the cell fates were thus sufficient information for quantification.

Interestingly, the humans and the machine learning approach agreed that patterns for larger $q$ are more easily recognizable which was indicated by a lower variability for the predictions for these $q$.

At present, this first approach of pattern quantification is limited to patterns that can be created by a simulation in which one parameter has a strong, predictable influence on the resulting pattern. We expect our approach to work equally well with data from other discrete pattern generation models that rely on one main parameter [27]. Extensions to more parameters by including more output nodes are an interesting aim for future work.

## A multilayer perceptron model allows accurate prediction of the fate of a given cell based on its direct neighbors

We showed that different to the pattern recognition task, prediction of the fate of a cell, based on its nearest neighbors, is possible in both 2D and 3D with MLPs. Our models were able to reconstruct not only single patterns, but all of them when trained with the whole simulated data set.

A difference appeared during the experiments with the simulated data between 2D and 3D. While the models with the 2D data needed at least the seven nearest neighbors as input, with the 3D data they needed the 12–14 nearest neighbors to achieve the best prediction. These numbers correspond to the typical number of cells that a given cell is in contact with in these tissues. Considering the details of our distance-based signaling model, this is an interesting finding. In the model, for larger $q$, there is an influence from cells further away. However, this is not essential for determining the fate of a given cell. The fate prediction works well just based on the nearest neighbors.

Crosstraining for the different $q$ showed that patterns for lower $q$ are very specific. For $q$ above 0.5, the data sets are more generally applicable to training for the different $q$. This matches our finding above that patterns for larger $q$ are more easily recognizable. Our pattern reconstruction problem is an inverse problem that is related to the wider field of image imprinting [28]. However, instead of the fixed arrangement of image pixels, we consider a

more general graph structure. Previous approaches for reconstruction of such general graphs [29] are supplemented by our approach. We introduce the notion of relying on a hypothesis for the pattern generation mechanism, namely that the direct neighborhood of a cell can determine the cell fate.

## Spatial statistics and pattern recognition suggest cell-cell interaction beyond the first neighbors in ICM organoids

Matching experimental data for the ICM organoids and simulations data based on Moran's index and pair correlation functions for six representative 48 h ICM organoids produced ambiguous results. We presented three examples that could be matched well based on pair correlation functions. The resulting $q$ varied between 0.2 and 0.8. The larger $q$ agree with our previous understanding from visual inspection that 48 h ICM organoids exhibit an engulfing pattern [5]. Matching the experimental data and the simulations based on Moran's index resulted in higher $q$ values. The discrepancies could arise because the pair correlation functions and the Moran's index highlight different aspects of the patterns, namely radial segregation and local clustering. In addition, the matching based on pair correlation functions was a visual comparison of curves. A more systematic approach might bring the values closer together.

Pattern recognition for the ICM organoid data based on the GNN provided an approach to assessing the pattern as a whole without the need for summary statistics. However, it was not straight forward as we did not have ground truth information. Therefore, we reverted to an approach based on synthetic data [30]. All *in vitro* organoids were predicted with dispersion parameters between 0 and 0.5. Although there were slightly different distributions between 24 h and 48 h organoids, even 48 h organoids were mostly predicted to have a low dispersion parameter. This suggests that 48 h ICM organoids show more clustering than 24 h organoids but are not fully segregated. This differs from our results from the spatial statistics approach that suggest higher values of $q$, i.e. longer range interactions.

The low dispersion parameter values agree with experimental evidence that suggests that intercellular communication via FGF4 in the stem cell cultures mainly couples nearest and second-nearest neighbors [8]. Previous modeling approaches for primitive endoderm and epiblast differentiation in mouse embryos however consider only nearest neighbor interactions [31].

## Pattern reconstruction shows a strong relationship between a cell's fate and its neighbors

Our pattern reconstruction approach shows that the nearest neighbor cells contain a lot of information. In over 70% of the cases, the fate of a cell can be predicted based on the information of neighboring fates. Interestingly, nine nearest neighbor cells are sufficient to achieve the best accuracy. The typical number of cells in contact with a given cell in ICM organoids is 14.

An accuracy of around 70% is lower than what we obtained for the pattern reconstruction of simulated patterns. A part of the drop off compared to the analysis for simulated data is expected due to the small amount of training sets and the change to four instead of two output classes. The test with only two classes showed an accuracy of 83.8% with Model3. To test this, we will need to extend our cell differentiation model to four categories. Unfortunately, developing a model with four steady states respresenting the different categories is not straight forward. Previous approaches have resulted in models with the three steady states u+v− and v+u− as well as u+v+ or u−v− [32, 33].

An additional factor could be the sensitivity of predictions for low $q$ on the training set. An interesting future experiment would be to separate the ICM data sets by the $q$s predicted from

the pattern recognition; followed by a subsequent training and testing for individual values of $q$. Unfortunately, for such a test, we would need more data sets.

The approach of developing machine learning models based on simulated data generated by already existing mathematical models shows great advantage. Since the simulated data have almost no limits in terms of their quantity and quality, machine learning models can be developed more quickly, and readily fitted to experimental data. Furthermore, by applying the machine learning models to simulated and experimental data, a comparison can be made between them to possibly draw conclusions for improvements of the mathematical models or generate new hypotheses for future experiments.

The combination of pattern simulation and pattern classification has also been used for *in vitro* generation of desired cell fate patterns [34]. So far, this approach has been restricted to bulls eye and multi-island patterns. Our results could contribute to expanding the range of possible patterns.

In summary, we were able to further our understanding of the biology of embryonic stem cell differentiation. The patterns in the ICM organoids agree with a mechanism that integrates signaling information from cells that are beyond the first neighbors. But still, the direct neighborhood is very informative as in all organoids it contains enough information to predict the fate of a given cell with more than 70% accuracy.

Hence, our pattern recognition and pattern reconstruction approaches go beyond mere pattern classification. Instead, the combination of mathematical modeling, simulation and deep learning allows linking the patterns directly to potential mechanisms. This approach is not limited to data from immunolabeling. It could be extended to address current challenges arising in spatial transcriptomics such as node classification and community detection including spatial imputation [11].

## Supporting information

**S1 Fig. PCF envelopes for 24 h organoids.** For each envelope, minimum and maximum of 1000 samples were used (see materials and methods for more details). Green borders highlight the organoids that show the most common pattern. Red borders highlight organoids, where cell type proportions are extremely skewed up to the point of no cells of one type. In blue, organoids with a similar pattern, that is not random, are highlighted. (Figure from [33]).
(TIF)

**S2 Fig. PCF envelopes for the ICM organoid data that developed for 48 h.** For each envelope, minimum and maximum of 1000 samples were used (see materials and methods for more details). Green borders highlight the organoids that show the most common pattern. (Figure from [33]).
(TIF)

**S3 Fig. Pattern for low dispersion parameters $q$ are more difficult to recognize.** Guesses for the dispersion parameter $q$ versus the correct value for $q$ for Model1.2D and data set A (Table 1).
(TIF)

**S4 Fig. Pattern reconstruction for 3D simulated organoids works best with 14 neighbors.** Accuracy of Model2.3D for data set D (Table 1) for different $q$ and different number of neighbors. Please note that for a clearer display, the y-axis starts at 40%.
(TIF)

**S5 Fig. Human predictions for pattern reconstruction show variability.** Accuracy of human predictions on simulated data set C (Table 1) by expert as well as the accuracy of Model1.2D and Model1.3D for the different values of $q$. Please note that for a clearer display, the y-axis starts at 35%.
(TIF)

**S6 Fig. Crosstraining for pattern reconstruction shows higher sensitivity to the training data for small $q$.** Model2.3D was trained on parts of data set D (Table 1) for the different values of $q$. Subsequently, these models were used to predict the cell fates in data sets split up by $q$. Each entry in the matrix corresponds to the accuracy of one such test run. The color coding ranges from dark red (worst accuracy) to dark green (best accuracy).
(TIF)

**S7 Fig. Normalized confusion matrices for pattern reconstruction for six ICM organoids.** Matrices show predictions plotted against true labels for three 24 h (left) and three 48 h (right) ICM organoids. The rows are normalized to 1.
(TIF)

**S1 Text. Details of the agent-based model and simulations for cell fate patterning in ICM organoids.**
(PDF)

**S2 Text. Details of the pair correlation functions for quantification of patterns of two cell types.**
(PDF)

## Acknowledgments

We thank Kerstin Schmid for critical reading of the manuscript.

## Author Contributions

**Conceptualization:** Robin Dirk, Jonas L. Fischer, Simon Schardt, Markus J. Ankenbrand, Sabine C. Fischer.

**Data curation:** Simon Schardt.

**Investigation:** Robin Dirk, Jonas L. Fischer, Simon Schardt.

**Methodology:** Robin Dirk, Jonas L. Fischer, Simon Schardt.

**Project administration:** Sabine C. Fischer.

**Software:** Robin Dirk, Jonas L. Fischer.

**Supervision:** Simon Schardt, Markus J. Ankenbrand, Sabine C. Fischer.

**Validation:** Robin Dirk, Jonas L. Fischer.

**Visualization:** Robin Dirk, Jonas L. Fischer.

**Writing – original draft:** Robin Dirk, Jonas L. Fischer.

**Writing – review & editing:** Robin Dirk, Jonas L. Fischer, Simon Schardt, Markus J. Ankenbrand, Sabine C. Fischer.

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
