## [Decision Letter · Decision Letter 0]

11 May 2023

Dear Prof. Dr. Fischer,

Thank you very much for submitting your manuscript "Recognition and reconstruction of cell differentiation patterns with deep learning" for consideration at PLOS Computational Biology.

As with all papers reviewed by the journal, your manuscript was reviewed by members of the editorial board and by several independent reviewers. In light of the reviews (below this email), we would like to invite the resubmission of a significantly-revised version that takes into account the reviewers' comments.

We cannot make any decision about publication until we have seen the revised manuscript and your response to the reviewers' comments. Your revised manuscript is also likely to be sent to reviewers for further evaluation.

Sincerely,

Jinyan Li

Academic Editor

PLOS Computational Biology

Daniel Beard

Section Editor

PLOS Computational Biology

Reviewer's Responses to Questions

**Comments to the Authors:**

Reviewer #1: The manuscript submission by Dirk et al describes a series of machine learning algorithms designed to investigate the characteristics associated with inner cell mass (ICM) organoids, which capture an early stage of blastocyst morphogenesis in cultured murine embryonic stem cells. The authors generate a large synthetic dataset of 2D and 3D cell configurations that describe binary state transitions of differentiation according to a “dispersion” parameter that accounts for influences on a cell ranging from only nearest neighbors to influences from cells further away and yields striking differences in multicellular patterning of the system. By generating data from thousands of organoids, a graphical neural network (GNN) model was trained and validated to predict the dispersion parameter value for a given image. The GNN predicts for actual experimental organoid images a range of dispersion values. Another ML model generates predictions of a cell phenotype (one of 4 classes based upon Nanog/Gata6 expression) solely based upon information regarding the closest neighbors’ phenotypes. While the model performs excellent for the synthetic dataset, accuracy of the experimental images is good but with considerable room for improvement. Primary claims in the paper are 1) algorithms can detect patterns that humans can’t; and 2) cell fate reconstruction can be predicted based upon a subset of the nearest neighbors.

The rigor and implementation of the machine learning in this paper is quite good. The use of a computational model yielded insights into the perception of cellular patterning detectable by ML versus human. However in the paper there are several critical shortcomings.

MAJOR:

1) Throughout the paper, the authors make claims (including the title) that the prediction of an individual cell’s phenotype from the neighboring cell information is pattern reconstruction. I find this misleading, one can think of this more as “spatial imputing” in which enough information about the surroundings allows for an estimate of what is happening at the center cell. That is quite different than pattern prediction which occurs on the macroscale.

As a sidenote, the motivation for the entire exercise is a bit weak. Why do we want to predict a single cell’s fate? Pivoting the emphasis toward imputation for spatial transcriptomics data acquisition could be an improvement to the paper.

2) The only description of the model is a reference to a pre-print that is not peer-reviewed. The authors need to provide enough description so that the readers can understand the nature of the model. Is it agent-based? Does the model contain any stochasticity or will the same initialized set yield the same outcome? Why is there variability in the organoid cell numbers for dataset B? How does the reader interpret the meaning of model-defined or experimentally extracted dispersion values?

3) Similarly, the authors never show any examples of the organoid images that they are working with. Statements in the paper such as “the spatial arrangement of epiblast and primitive endoderm precursor cells is non-random but visually unrecognizable” should be backed up by these images. The authors should show an example of where inaccurate predictions occur spatially on the organoid.

4) The estimation of cell fate by both machine and expert is made from cell state and rank order distance from the predicted cell. All spatial information is lost and I don’t understand the choice in discarding this. For example, in Figure 5, if the user knew that all the red cells were clustered to the left side of the cell in question, and the blue cells were clustered to the right, that information has both biological implications (e.g. a larger localized buildup of morphogen) and implications for accuracy.

5) The authors’ choice in binning their cell data does not normalize expression levels by cell area, and subsequent analysis is based upon the centroid coordinates. How are the results impacted if normalization is performed, which will change the distribution of N-G-, N+G-, N+G+, N-G+ cells?

6) Why do bigger, older organoids have higher dispersion values? This result is not sufficiently discussed.

7) I do not understand the conclusion drawn by the sentences lines 444-447: “For the ICM organoid data, we obtained an accuracy of more than 70% for nine neighbors, which is less than the typical 14 cells that are directly in contact with a given cell. This is in agreement with a short range cell-cell communication as predicted by the pattern recognition”.

Why does having less than 14 cells needed make this a short-range process? If all the cells in the vicinity are of similar phenotype (i.e. a cluster), then the immediate cells are reflecting the distant cells, this says nothing about the mode of communication.

MINOR:

8) Throughout the text are references to “2D organoids”. These are 2D colonies.

9) The justification for discarding organoids with > 2/3 of one type is weak. How many organoids fell into this category? The radially organized organoids that appear at this stage may have a central core of less than 1/3 and highly relevant to the morphogenesis under investigation

Reviewer #2: Dirk et al combine machine learning (ML) algorithms and mathematical models to analyze spatial patterns of cellular fate decisions. More specifically, they assume a mathematical model of cellular fate decision where cells can interact with their neighbors and adopt two different fates. In this model, a parameter, q, sets the range of cell-cell interaction. Then, they train ML algorithms on patterns of cell fate decision generated in silico via their model to predict the parameter q, given a specific spatial distribution of cell fates (what they call "pattern recognition" task). Additionally, they train ML algorithms to predict the fate of a cell from the fates of the neighboring cells ("pattern reconstruction" task).

In doing so, they use simulated and imaging data generated from mouse ICM organoids. When applying their model to mouse ICM organoids data, they conclude that the analysis suggests the existence of a short-range cell-cell communication controlling cellular fate decision, in agreement with some experimental evidence.

The approach used by the authors (the combination of ML with mathematical modelling) is very interesting, and very much needed in biology, where there's an increasing availability of large-scale datasets from which we can extract potentially interesting patterns. However, we're still largely unable to obtain valuable insights into the molecular interactions that generate them. The combination of approaches from ML and mathematical modelling can help address this issue.

Overall the implementation of the models is sound, and the manuscript is well-written.

However, I have some points that should be addressed before publication.

1. Can one predict the value of q (in both 2D and 3D) by computing some spatial statistics used in marked point processes, such as the Stoyan's mark correlation function, the mean or variance-mark function, etc.? Before using more complex approaches based on deep learning, The authors should show that more straightforward strategies like this do not work or are less accurate. This would be more compelling than comparing the accuracy of the deep learning algorithms with the manual annotation of human experts.

2. For some values of q, the authors observed a decrease in the model performance in the task of pattern reconstruction. For example, in 2D using Model 2, the testing accuracy was below 80% for higher values of q (line 355), where engulfing patterns are obtained. Does the performance improve if, in addition to the cell fate and distances of the nearest neighbors, the cell's position is also given as input (e.g., using some sort of distance from the "boundary")? It seems like, especially in some situations (like in the engulfed patterns), the relative position of the cell might also be predictive of its fate. It would be interesting to see whether the addition of the cell position can improve the models' accuracy with the in vitro data.

3. Given that 4 categories are used with the in vitro data, the authors should generate in silico data with 4 categories as well (instead of 2) and then compare the accuracy they obtain with the in silico vs the in vitro data. This would also be useful to support their claim that the lower accuracy observed with the in vitro data is (partly) due to the greater number of categories.

**Have the authors made all data and (if applicable) computational code underlying the findings in their manuscript fully available?**

Reviewer #1: Yes

Reviewer #2: None

PLOS authors have the option to publish the peer review history of their article (what does this mean?). If published, this will include your full peer review and any attached files.

Reviewer #1: No

Reviewer #2: No
---

## [Decision Letter · Decision Letter 1]

29 Aug 2023

Dear Prof. Dr. Fischer,

Thank you very much for submitting your manuscript "Recognition and reconstruction of cell differentiation patterns with deep learning" for consideration at PLOS Computational Biology. As with all papers reviewed by the journal, your manuscript was reviewed by members of the editorial board and by several independent reviewers. The reviewers appreciated the attention to an important topic. Based on the reviews, we are likely to accept this manuscript for publication, providing that you modify the manuscript according to the review recommendations.

Sincerely,

Jinyan Li

Academic Editor

PLOS Computational Biology

Daniel Beard

Section Editor

PLOS Computational Biology

Reviewer's Responses to Questions

**Comments to the Authors:**

Reviewer #1: The authors have done an excellent job revising the manuscript to address concerns. The addition of spatial autocorrelation metrics greatly enhances the paper. I found the interactive web visualization very useful for exploring the data. I do have lingering comments that were not fully addressed by the edits:

1) The referencing of 2D systems as organoids still appears throughout the document, e.g. Figure 2, lines 229, 368

2) Lines 323-325: “We find that 24 h after organoid formation, the Moran's index is consistently larger than 0, indicating a non-random distribution of the cells (Fig. 7). For 48 h organoids, the average Moran's index increases further, indicating stronger clustering of the two cell types.”

It is unclear which dataset is being referred to here. It would be helpful to explicitly label the dataset used for this observation. Is it experimental images or simulations?

3) Unfortunately, despite the addition of Figures 8, 9, 10, S1, S2, there is an unclear degree of overlap in results from other publication. Whatever has been accepted for publication must be quite different from the arxiv link. How much of what is being presented is new analyses on data presented in the other paper versus reproductions of the same figures from that paper? Typically, with a large degree of overlap, a journal will request the accompanying pre-print to make this determination.

4) I find figure 10 very hard to interpret. I understand that the attempt is to show “good” versus “bad” organoids modeled at different q values, however the use of opacity in the image does not make sense to me. Am I to interpret this as only organoid 36 is well-captured by the Moran’s index statistic? What are the implications of this result?

Reviewer #2: The authors have addressed all the points that I've raised, and I have no further comments.

**Have the authors made all data and (if applicable) computational code underlying the findings in their manuscript fully available?**

Reviewer #1: Yes

Reviewer #2: None

PLOS authors have the option to publish the peer review history of their article (what does this mean?). If published, this will include your full peer review and any attached files.

Reviewer #1: No

Reviewer #2: No

Figure Files:

Data Requirements:

Reproducibility:

References:

---

## [Decision Letter · Decision Letter 2]

9 Oct 2023

Dear Prof. Dr. Fischer,

We are pleased to inform you that your manuscript 'Recognition and reconstruction of cell differentiation patterns with deep learning' has been provisionally accepted for publication in PLOS Computational Biology.

Best regards,

Jinyan Li

Academic Editor

PLOS Computational Biology

Daniel Beard

Section Editor

PLOS Computational Biology

Reviewer's Responses to Questions

**Comments to the Authors:**

Reviewer #1: The authors have done a great job of addressing my last concerns.

**Have the authors made all data and (if applicable) computational code underlying the findings in their manuscript fully available?**

Reviewer #1: Yes

PLOS authors have the option to publish the peer review history of their article (what does this mean?). If published, this will include your full peer review and any attached files.

Reviewer #1: No

---

## [Editor Report · Acceptance letter]

24 Oct 2023

PCOMPBIOL-D-22-01868R2 

Recognition and reconstruction of cell differentiation patterns with deep learning

Dear Dr Fischer,

I am pleased to inform you that your manuscript has been formally accepted for publication in PLOS Computational Biology. Your manuscript is now with our production department and you will be notified of the publication date in due course.

With kind regards,

Anita Estes
